# A numerical modeling investigation of the role of diabatic heating and cooling in the development of a mid-level vortex prior to tropical cyclogenesis. Part I: The response to stratiform components of diabatic forcing

5  Melville E. Nicholls[1], Roger A. Pielke Sr.[1], Donavan Wheeler[2], Gustavo Carrio[3], and Warren P. Smith[1]

[1]Cooperative Institute for Research in Environmental Sciences, Department of Atmospheric and Oceanic Sciences, University of Colorado, Boulder, CO 80309, USA
[2]Department of Atmospheric and Oceanic Sciences, University of Colorado, Boulder, CO 80309, USA
[3]Spire Global, Inc, Boulder, Colorado, USA

*Correspondence to*: Melville E. Nicholls (Melville.Nicholls@colorado.edu)

**Abstract.** Mid-tropospheric mesoscale convective vortices have been often observed to precede tropical cyclogenesis. Moreover, recent cloud resolving numerical modeling studies that are initialized with a weak cyclonic mid-tropospheric vortex sometimes show a considerable intensification of the mid-level circulation prior to the development of the strong

cyclonic surface winds that characterize tropical cyclogenesis. The objective of this two-part study is to determine the processes that lead to the development of a prominent mid-level vortex during a simulation of the transformation of a tropical disturbance into a tropical depression, in particular the role of diabatic heating and cooling. For simplicity simulations are initialized from a quiescent environment. In this first part, results of the numerical simulation are described and the response to stratiform components of the diabatic forcing is investigated. In the second part, the contribution of

diabatic heating in convective cells to the development of the mid-level vortex is examined.

Results show that after a period of intense convective activity, merging of anvils from numerous cells creates an expansive stratiform ice region in the upper troposphere, and at its base a mid-level inflow starts to develop. Subsequently conservation of angular momentum leads to strengthening of the mid-level circulation. A twelve-hour period of mid-level vortex intensification is examined during which the mid-level tangential winds become stronger than those at the surface. The main

method employed to determine the role of diabatic forcing in causing the mid-level inflow is to diagnose it from the full physics simulation and then impose it in a simulation with hydrometeors removed and the microphysics scheme turned off. Removal of hydrometeors is achieved primarily through artificially increasing their fall speeds three hours prior to the twelve-hour period. This results in a state that is in approximate gradient wind balance, with only a weak secondary circulation. Then, estimates of various components of the diabatic forcing are imposed as source terms in the thermodynamic

equation in order to examine the circulations that they independently induce. Sublimation cooling at the base of the stratiform ice region is shown to be the main factor responsible for causing the strong mid-level vortex to develop, with smaller contributions from stratiform heating aloft and low level melting and evaporation. This contrasts with the findings of

previous studies of mid-latitude vortices that indicate sublimation plays a relatively minor role. An unanticipated result is that the central cool region that develops near the melting level is to a large degree due to compensating adiabatic ascent in response to descent driven by diabatic cooling adjacent to the central region, rather than in situ diabatic cooling. The midlevel inflow estimated from stratiform processes is notably weaker than for the full physics simulation, suggesting a moderate contribution from diabatic forcing in convective cells.

## 1 Introduction

There are numerous observational studies showing that mesoscale convective vortices (MCVs) that form at mid-levels in the stratiform regions of tropical disturbances are often precursors to tropical cyclogenesis (e.g. Zehr 1992; Harr and Elsberry 1996; Simpson et al. 1997; Bister and Emanuel 1997; Ritchie and Holland 1997; Reasor et al. 2005; Raymond and López Carrillo 2011; Gjorgjievska and Raymond 2014). Recent cloud resolving simulations that are initialized with a relatively weak vortex with maximum winds at mid-levels over a warm ocean surface and for a non-sheared environment suggest that there may be two distinct pathways to tropical cyclogenesis that can occur even for this very idealized posing of the tropical cyclogenesis problem (Nicholls and Montgomery 2013; hereafter NM13). One of the pathways showed an evolution similar to earlier studies conducted by Hendricks et al. (2004) and Montgomery et al. (2006; hereafter M06), which suggests a relatively incidental role of the mid-level vortex. Small-scale cumulonimbus towers possessing intense cyclonic vorticity in their cores emerge as the preferred coherent structures. The term vortical hot towers (VHTs), has been coined for these rotating cumulonimbus that occur in vorticity-rich environments (Hendricks et al. 2004). The diabatic heating in the cores of numerous VHTs collectively drove a quasi-balanced system scale transverse circulation that at low levels converged cyclonic vorticity of the MCV and small-scale vorticity anomalies generated by subsequent tower activity. This resulted in a spin-up of the near surface tangential winds and a relatively gradual decrease in the radius of maximum surface winds. While there was a system scale mean radial inflow in the mid troposphere that resulted in a relative maximum in the mid-level tangential winds of the M06 control simulation, this did not appear to play any obvious role in the intensification of the low level circulation.

The second pathway, termed pathway Two, that occurred in the simulations of NM13 showed similarities with the results of Nolan (2007). There was a significant strengthening of the mid-level circulation and typically a contraction in size followed by the sudden formation of an intense small-scale vortex with strongest winds at the surface near the centre of the large-scale circulation. This subsequently became the core of an intensifying tropical cyclone. Nolan (2007) hypothesized that the intensification of the mid-level vortex led to a large increase in the efficiency of the conversion of latent heat energy to the kinetic energy of the cyclonic wind field, which is in line with the theoretical studies of Schubert and Hack (1982) and Vigh and Schubert (2009). Thus the development of the mid-level vortex was seen as playing an integral role in the transformation of a tropical disturbance into a tropical cyclone. NM13 postulated that another consideration was that the cold anomaly at low-levels associated with the mid-level vortex caused a decrease of the low-level static stability and therefore

favoured convection to develop at low-levels near the centre. For their simulations the typical scenario involved a low-level vorticity anomaly that had formed some distance from the centre spiralling inwards and becoming an intense vortex at the centre. Concurrent with the development of the central intense vortex was persistent convection usually located at the edge of a surface cold pool. NM13 hypothesized that the main cause of the mid-level inflow in their simulations that resulted in the spin up of the mid-level cyclonic circulation was sublimation at the base of the extensive sloping ice layer that developed prior to tropical cyclogenesis.

A notable earlier work that investigated the role of a mid-level mesocyclone during tropical cyclogenesis was carried out by Bister and Emanuel (1997). They analysed the genesis of Hurricane Guillermo (1991) in the eastern Pacific Ocean and moreover used an axisymmetric numerical model to determine how a pre-existing mid-level vortex might promote the development of an enhanced surface circulation, which could then lead to tropical cyclogenesis. A rainshaft was prescribed within an initial mid-level vortex and it was found that evaporative cooling of the raindrops led to a gradual descent of the vortex towards the surface. Following this period of low-level cooling the increased surface winds produced stronger surface fluxes, which eventually led to an increase of low-level equivalent potential temperature that promoted new convection and hence the strengthening of the surface circulation. However, this theory does not seem to be borne out by recent cloud-resolving simulations since they do not show such a significant expansion downwards of the mid-level vortex with time. For instance, for the NM13 simulations the height of the maximum cyclonic winds of the vortex typically descended by only 1-2 km prior to genesis occurring. It may be that the vortex builds downwards with time in the NM13 simulations due to the peak diabatic cooling shifting to lower levels. A possible explanation is that sublimation, and to a lesser extent evaporation, of falling hydrometeors, which appear to be primarily responsible for the mid-level inflow, results in increasing humidity at mid-levels, so that subsequent sublimation and evaporation are reduced allowing more hydrometeors to reach lower levels as time progresses, thereby shifting the peak diabatic cooling and horizontal convergence downwards. Bister and Emanuel (1997) give a similar explanation for how evaporation is enhanced at lower levels with time in their simulation. This possible mechanism for a mid-level vortex building downwards is not the same thing as saying there is a downward transport of vertical vorticity. A net downward transport of vorticity from the mid-level vortex is not consistent with the theorems of Haynes and McIntyre (1987) as pointed out by Raymond et al. (2011) and Kilroy et al. (2018). They emphasize that vertical vorticity enhancement in the mid-level vortex fundamentally involves layerwise concentration.

Based on observational analyses Simpson et al. (1997) and Ritchie and Holland (1997) have hypothesized that strengthening of the midtropospheric cyclonic circulation could occur through the merger of mid-level mesoscale vortices and at the same time this would enhance the cyclonic circulation in the boundary layer. This was then followed by more active development at low levels, although the precise nature of this development and its relation to the stronger mid-level circulation was difficult to ascertain and was said to merit further study.

Another theory is based on the idea that the thermodynamic environment associated with the mid-level vortex is conducive to convection with a more bottom-heavy mass profile (Raymond et al. 2011; Raymond and López Carrillo 2011; Gjorgjievska and Raymond 2014, Davis and Ahijevych 2012). Such a vertical mass flux profile would be expected to result

in strong low-level convergence of mass and vorticity and spin up of a low-level cyclone. However, a tropical cyclogenesis simulation that included the ice phase by Kilroy et al. (2018) did not develop a bottom heavy mass flux, which if true in general would have implications for this theory..

Davis and Ahijevych (2012) studied three tropical weather systems that occurred during the Predepression Investigation of Cloud Systems in the Tropics (PREDICT), outlined in Montgomery et al. (2012). The two developing cases featured a faster increase of the midtropospheric circulation prior to genesis compared to the lower-tropospheric circulation. For the non-developing case the presence of significant environmental vertical wind shear forced a considerable misalignment of the midtropospheric and lower tropospheric circulation centres. This allowed dry air to intrude above the low tropospheric centre, which limited the development of deep convection and prevented the strengthening of the vortex. While there was some misalignment of the midtropospheric and lower tropospheric circulation centres for the developing cases, alignment eventually occurred and was concurrent with quasi-persistent deep moist convection and tropical cyclogenesis occurring. The exact processes responsible for causing the vertical alignment were not however able to be assessed from the available data set.

Dunkerton et al. (2009) proposed the "marsupial paradigm" that provides a theoretical framework for understanding tropical cyclogenesis in easterly waves. The Kelvin cat's eye within the critical layer, or "wave pouch", was identified as a favourable environment for tropical cyclogenesis. This was followed by several numerical modeling studies having the objective of evaluating the theory (Montgomery et al. 2010; Wang et al. 2010a and b, 2012). These studies support the conclusion that the quasi-closed circulation in a reference frame moving with the wave is a region that favours persistent deep convection, vorticity aggregation, and moistening, which increases the likelihood of tropical cyclogenesis occurring. Their perspective was that mid-level vortices probably were not very relevant to the Atlantic and Eastern Pacific sectors outside the Intertropical Convergence Zone (ITCZ), although they could be playing a more prominent role in the Western Pacific. On the other hand the PREDICT study of Atlantic systems by Davis and Ahijevych (2012) showed a significant misalignment of the mid-level and surface circulations, sometimes as much as 200 km or more. Also Hurricane Guillermo (1991) in the Eastern Pacific investigated by Bister and Emanuel (1997) was observed to have a mid-level vortex, so this issue may not be completely resolved as of yet.

Recently Kilroy et al. (2018) carried out a numerical modeling study that led them to conclude that the presence of a mid-level vortex that developed when the ice phase was activated did not play an essential dynamical or thermodynamical role in the genesis process. They found that a simulation with warm rain microphysics underwent genesis much more rapidly than one with the ice phase included, which agrees with the results of NM13. However their warm rain simulation, which did not form a prominent mid-level vortex underwent a sudden jump in the radius of maximum winds at low-levels as a very small vortex formed, rather than showing a gradual decrease in the radius as occurred in the NM13 warm rain simulation. The development of a prominent mid-level vortex did seem to be required in the NM13 simulations prior to the sudden formation of a very small surface concentrated vortex. While many numerical modeling studies are indicating that mid-level vortices are not essential for tropical cyclogenesis to occur, there are reasons to believe that when they form they can exert significant

influences on the subsequent evolution of a tropical disturbance. For instance, by causing dry air advection over the low-level circulation as indicated by the study of Davis and Ahijevych (2012) discussed above. We have also conducted simulations in weak vertical wind shear that suggest if a strong mid-level vortex forms it can subsequently be the focal point for the development of new convection, which may be due to enhanced humidity in the vortex and decreased low-level stability (unpublished). Moreover, if the centre of a mid-level vortex is displaced relative to the centre of the low-level circulation this can result in strong local vertical wind shear between the two centres, which could be expected to influence convective cell dynamics.

Penny et al. (2016) conducted a numerical modeling investigation of a non-developing tropical disturbance, and experimented with various microphysical schemes. Several simulations overdeveloped and produced a tropical cyclone unlike the observed system. The fastest development occurred for a case that only had warm rain microphysics, which is in agreement with the results of NM13 and Kilroy et al. (2016). Interestingly, the simulation that agreed best with observations and didn't develop used a relatively simple microphysics scheme. Comparison with a simulation that used a more complex scheme showed that the case that didn't develop produced more snow aloft, whereas the overdeveloping case produced a lot of graupel that fell out quickly.

Results of cloud-resolving numerical simulations are showing a strong sensitivity to the representation of microphysical processes, and if conditions are conducive to produce an extensive stratiform canopy aloft and develop a mid-level vortex, which can significantly effect the rate of development, it appears crucial to realistically represent these processes if improvement to operational numerical model forecasting is to be achieved. This manuscript focuses on the problem of how mid-vortices initially form, and it is well to keep in mind this is only an analysis with a particular configuration of one model (The Regional Atmospheric Modeling System: RAMS). An earlier version of this model did not produce significant mid-level vortices (M06), while a later version did (NM13). So there is a lot of uncertainty about whether this model, or any other model for that matter, is faithfully representing reality. We hope that a detailed study of the specific physical processes taking place in this simulation will add to the discussion on this subject, and contribute to understanding the complicated mechanisms responsible for causing mid-level vortices to form.

A summary of the current understanding of how mid-level vortices form in many mesoscale convective systems (MCSs) is provided in Houze (2004). Some mid-level vortices form in the trailing stratiform region of linear convective elements, such as squall lines, whereas in other cases the convection does not have a linear organization. For the particular situation simulated in this study, which has an initial condition consisting of a pre-existing weak mid-level vortex in a quiescent environment, a fairly axisymmetric stratiform region forms aloft that is punctuated by numerous deep and transient convective cells. Therefore the stratiform cloud for this situation is not clearly separated in space from a convective region. While the convective cells have a large degree of randomness in their spatial distribution, there is a definite tendency for them to be more concentrated near the radius of maximum surface winds where surface sensible and latent heat fluxes are largest and also where frictional convergence is substantial.

There have been several numerical modeing studies that focus on the development of MCVs at mid-latitudes (Zhang

1992; Chen and Frank 1993; Rogers and Fritsch 2001; Conzemius and Montgomery 2009; Davis and Galarneau 2009; Wang et al. 2013). These studies have not emphasized a particularly strong role for sublimation in causing mid-level vortices to form, with the exception of Zhang (1992) who found that cooling in the trailing stratiform region of a squall line appeared to play a dominant role in causing a mesoscale vortex to form and sublimation as well as melting and evaporation was

5 responsible for the cooling. Also, some modeling studies of mid-latitude squall lines indicate that sublimation cooling occurs in the rear inflow air as it begins to descend under the cloud deck (Yang and Houze 1995; Braun and Houze 1997). Overall the contribution of sublimation in causing MCVs to form at mid-latitudes appears to be considered relatively minor based on these past studies with a greater emphasis being placed on latent heating in the stratiform cloud layer aloft and melting and evaporation at lower levels. On the other hand, the study by NM13 suggests sublimation may have a major role in causing

MCVs to form in tropical disturbances.

The main goal of this study is to demonstrate that the development of a strong mid-level vortex in the numerical simulations of NM13 is indeed primarily caused by mid-level inflow driven by diabatic cooling at the base of the stratiform ice layer. Secondly we will examine the contribution of other potential forcing mechanisms, in particular stratiform heating aloft and low-level cooling, and compare them with one another. For this purpose a high-resolution numerical simulation

with RAMS was conducted that shows evolution along pathway Two, and a twelve-hour period during which a prominent mid-level vortex began to form chosen for detailed analysis. The main approach taken in this study to ascertain the effects of diabatic heating on the wind fields is to examine its spatial distribution and evolution in time over the twelve-hour period and then impose simplified functions approximately representing various stratiform components of this diabatic heating as source terms within the thermodynamic equation for simulations without microphysics activated. These diabatic heating functions

were imposed on an initial state that was close to gradient wind balance and similar to the primary circulation in the full physics simulation at the beginning of the twelve-hour period. Differences between the strength of the mid-level inflow and the structure of the mid-level vortex produced in the idealized simulations with stratiform heating functions and the full physics simulations are identified and it is inferred that some of these differences are due to convective scale processes.

An outline of this manuscript is as follows: In section 2 the numerical model and the initial conditions for the simulation

are described. Section 3 discusses the methodology, which introduces the formula used to diagnose the latent heating and cooling rates. This section also outlines the method used to produce a near gradient wind balanced state. Results are presented in section 4, which begins by discussing the full physics simulation and in particular those aspects important for estimating the diabatic forcing functions. The following subsection describes the near gradient balanced state. The next subsection presents the estimated forcing functions and results of imposing them in simulations initiated with the near

gradient wind balanced state. Finally conclusions are presented in section 5.

## 2 Numerical model and experimental design

RAMS is a nonhydrostatic numerical modeling system comprising time-dependent equations for velocity, nondimensional pressure perturbation, ice-liquid water potential temperature, total water mixing ratio, and cloud microphysics, developed at Colorado State University (Pielke et al. 1992; Cotton et al. 2003). The microphysics scheme has categories for cloud droplets, rain, pristine ice crystals, snow, aggregates, graupel and hail (Walko et al. 1995). The model has the capability of using multiple nested grids, which allows explicit representation of cloud-scale features within the finest grid while enabling a large domain size to be used, thereby minimizing the impact of lateral boundaries. Further details of the model physics used for this tropical cyclogenesis simulation can be found in Nicholls (2015). Pertinent to this study is the use of the two-moment version of the microphysics scheme, which predicts the number concentration of hydrometeors and enables the mean diameter to evolve (Meyers et al. 1997), and a binned cloud-droplet riming scheme (Saleeby and Cotton 2008). For this simulation radiation was not included.

The simulation uses three grids with horizontal grid increments of 12, 3 and 1 km, and (x, y, z) dimensions of 170×170×48, 202×202×48, 302×302×48, respectively. Each finer scale grid is centred within the next coarsest grid. The vertical grid increment is 60 m at the surface and gradually stretched with height to the top of the domain at $z = 22.3$ km. A Rayleigh friction layer is included above $z=15.3$ km to damp upward propagating gravity waves.

Similarly to M06 and NM13, the temperature structure is the mean Atlantic hurricane season sounding of Jordan (1958). The initial moisture profile is the same as used for the reduced Convective Available Potential Energy (CAPE) experiments in M06, referred to as B2 and B3 in that study. This profile has low level moisture reduced from the Jordan sounding by a maximum of 2 g kg$^{-1}$ at the surface. The reduced moisture of this profile may be more representative of the environment surrounding a tropical cyclone, which would be expected to have a moister core. It also has the advantage of producing a more focused development, so that there is less convective activity in the coarser grids that only poorly resolve convective scales.

The equations for the initial MCV are given in M06 and portrayed in NM13 (Fig. 1 of that paper). Maximum tangential winds are at a height of 4 km above sea level and at a radius of 75 km. At the radius of maximum winds (RMW) cyclonic velocities are 8 m s$^{-1}$ at $z=4$ km and 4 m s$^{-1}$ at the surface. The initial vortex is moistened below 8 km and for a radius less than the RMW, to 85% of saturation with respect to liquid. This moisture anomaly is linearly reduced to environmental values from the RMW to a radius of 25 km beyond the RMW.

## 3 Methodology

The numerical model uses ice-liquid water potential temperature as a prognostic thermodynamic variable, which is conserved under vapour to liquid, vapour to ice and liquid to ice phase changes (Tripoli and Cotton 1981). Consequently, the diabatic heating rate due to phase changes of water substance is not a variable explicitly used by the model, and not

straightforward to calculate. In this study the net diabatic heating due to phase changes of water substance is diagnosed from the formula:

$$\delta Q_m = L_c \delta r_l + (L_c + L_f)\delta r_i \,, \tag{1}$$

where $\delta Q_m$ is the net latent heat released or absorbed per unit mass in a model time step within a grid volume, $\delta r_l$ is the change in the mixing ratio of liquid, $\delta r_i$ is the change in the mixing ratio of ice, $L_c$ the latent heat of condensation and $L_f$ the latent heat of fusion. The derivation of Eq. 1 is given in Appendix A. During a model time step the changes in mixing ratios of liquid and ice resulting from vapour diffusion, melting etc. are calculated in the microphysics algorithm prior to sedimentation. The amount of heat released or absorbed within a model time step determined from (1) is used to calculate a heating rate per second $Q_m$.

To examine the dynamical response to imposed latent heating or cooling the diagnosed diabatic forcing is added as a source term to the RAMS prognostic equation for ice-liquid water potential temperature, which in the absence of hydrometeors is simply potential temperature. The source term is given by:

$$S = \frac{Q_m \theta}{c_p T} \tag{2}$$

where $\theta$ is potential temperature, $c_p$ is the specific heat capacity at constant pressure and $T$ is temperature.

A simulation was conducted that evolved along pathway Two and a twelve-hour period during which the prominent mid-level vortex began to develop was chosen for analysis. The diabatic heating over this twelve-hour period was examined and idealized axisymmetric functions representing three stratiform components were constructed: (1) Mid-level cooling beneath the stratiform ice region due to sublimation; (2) Stratiform heating aloft due to vapour deposition onto ice crystals; (3) Stratiform cooling at low levels due to melting and evaporation.

Obviously, the estimated diabatic forcing cannot be imposed while there is microphysics activated, since the latent heating would be double-counted. The best approach would seem to be to impose the diabatic forcing on a state that is initially close to gradient wind balance, which has had the hydrometeors removed and no longer contains strong convective scale motions, or radial flows; with the proviso that this state should still be reasonably similar to the actual state occurring in the full physics simulation at the beginning of the twelve hour period in terms of the primary wind circulation, and temperature and moisture fields. Some justification for this approach is found in M06, which compared the results of the cloud model simulation with the quasi-balanced Sawyer-Eliassen model (Eliassen 1951) during the period cyclogenesis occurred. The Eliassen equations assume that the axisymmetric motions evolve in hydrostatic and gradient wind balance. Reasonable agreement between the balance model's solution when forced with estimated diabatic heating rates, and the cloud model were obtained, suggesting that on the whole the evolving vortex remains fairly close to gradient wind balance. Therefore, it might be surmised that if the hydrometeors can be removed and the convective motions allowed to decay, then the vortex will adjust to a state that is virtually in gradient wind balance, and this state will not be too different from the quasi-balanced

state at the beginning of the period of interest. Consequently, the sub-problem arises of how to remove the hydrometeors and allow the convection and transverse circulation to decay without causing a large change to the balanced part of the flow. The best approach we found was to gradually allow the convection to decay by artificially increasing the fall velocity of hydrometeors, slightly drying out the boundary layer, and turning off the sea surface fluxes for several hours prior to the twelve-hour period of interest.

After determining the twelve-hour period that would be analysed the model was started from a history file three hours earlier and then run for three hours with the following modifications in order to achieve a near gradient wind balanced state: The increase of the fall speed was achieved by setting the vertical displacement in the sedimentation scheme to the maximum allowable within a model time step. By half-an-hour this resulted in most of the hydrometeors being removed. In addition a water vapour sink was applied within the boundary layer and surface fluxes were turned off to impede the development of convection. A history file from this run was then used to initialize simulations with imposed diabatic forcing functions and with the microphysics scheme deactivated.

The impact of stratiform precipitation drag or waterloading was also evaluated to determine if it might be a significant factor. In RAMS it is represented in the equation for vertical velocity by the term $-g(r_c+r_r+r_p+r_s+r_a+r_g+r_h)$ where g is the gravitational constant and the terms in brackets are the mixing ratios of cloud water, rain, pristine ice, snow, aggregates, graupel and hail categories, respectively. The stratiform mixing ratio of the total hydrometeors can be estimated from an azimuthal average and then applied as a forcing term to the vertical velocity equation in a simulation without the microphysics scheme activated similarly to the approach taken to examine the response to diabatic forcing.

Five main experiments were conducted altogether and are listed in Table 1. They include simulations that examine the effects of the three diabatic forcing functions imposed independently, a simulation that investigates their combined effect, and a simulation to determine the impact of precipitation drag.

## 4 Results

### 4.1 General description of the simulation

Figure 1 shows time series of several variables for the first 108 h of the simulation during which tropical cyclogenesis occurred. The system evolved along pathway Two with low-level winds strengthening initially (Fig. 1a and b), followed by the formation of a significant mid-level vortex at 58 h (Fig. 1c) and then the sudden development of very small surface concentrated vortex at 90 h (Fig. 1d). For this study the twelve-hour period between 48-60 h was chosen for analysis since this is when the mid-level vortex first became prominent. During this period it can be seen that there was a notable increase in the total ice content with a peak occurring at 57 h (Fig. 1e), Moreover the upward convective mass flux (defined here as having vertical velocities $>0.5$ m s$^{-1}$) at the 3 km level shows a strong peak at 55 h (Fig. 1f).

Figure 2 shows horizontal sections of vertical velocity and mixing ratio of hydrometeors at z=9.5 km, at 48 h, and 60 h in the fine grid. At 48 h several small intense convective cells can be identified with vertical velocities in excess of 10 m s$^{-1}$,

most of them between 25-50 km from the centre. The hydrometeor field at this time looks like a ragged shaped annulus with considerable asymmetry, where there are peak values in small convective cells, and larger stratiform areas of weaker values that are remnants of anvils from numerous earlier cells that have merged. At 60 h there has been a slight increase in the number of cells and they tend to be closer to the centre. The area of stratiform hydrometeor content has grown considerably
in this twelve-hour period.

    Figure 3 shows height/radius sections of azimuthally averaged fields of radial velocity, tangential velocity, total hydrometeor mixing ratio, potential temperature perturbation, vertical velocity and diabatic heating, at 48 h. There is a weak mid-level inflow with a height that decreases as the radius decreases, and also a weak inflow near the surface. The tangential wind field shows there has been an increase of the low level circulation from initial values with wind speeds reaching 10 m s$^{-1}$
at z=1 km, r=55 km. Moreover, there is a maxima at z=6.5 km, r=35 km. Clearly this mid-level maximum of tangential winds is likely to be associated with the mid-level inflow. The hydrometeor mixing ratio has large values at the radius of the intense convective cells shown in Fig.2b. The melting level is at a height of 4.7 km so the mid-level inflow can be seen to be above this and at the base of the sloping stratiform ice layer. The perturbation potential temperature field shows a notable central warming between 8-10 km. The low-level cooling seen between r=40 to 120 km is from the initial vortex, which still
has an effect on the temperature field even after 48 h. There has been low level cooling near the centre due to convective scale downdrafts. There is also a thin layer of cooling at z=13 km. The vertical velocity shows a top-heavy profile with a maximum at z=9.5 km, r=35 km. The diabatic heating profile is also top-heavy with a maximum at z=8.5 km, r=35 km, and interestingly a secondary maximum above at z=10.5 km. It is possible that the secondary maximum is due to supercooled cloud droplets in the rising updraft freezing at upper levels where temperatures are very cold. Homogeneous and
inhomogeneous production of ice crystals is a strong function of temperature and becomes large below -35 °C, which is at approximately a height of 10 km for the basic state sounding. Weak diabatic cooling can be seen to occur at the base of the sloping stratiform ice region at radii where deep convection is absent.

    Figure 4 shows height/radius sections of azimuthally averaged fields at 60 h. There has been a very significant increase in the strength of the mid-level inflow from twelve hours earlier, with velocities now reaching a magnitude of 3-4 m s$^{-1}$. Also
there is a significant outflow aloft. There has been a slight increase in the magnitude of the low-level inflow. The tangential velocity now shows maximum values at mid levels of 12 m s$^{-1}$, and there has been some increase of the low-level circulation. Weak diabatic cooling occurs at the base of the sloping stratiform ice region. Hydrometeor mixing ratios have increased significantly and the stratiform region aloft has expanded laterally. There has been considerable central warming at upper levels and central cooling between z=3-5 km. There is also a notable cool region at the base of the stratiform ice layer
Vertical velocities have increased in magnitude and still have a top heavy profile. The diabatic heating in convective cells increased as well as the cooling at the base of the stratiform region.

    Figure 5 shows the time and azimuthally averaged diabatic heating field for the twelve-hour period. The time averaged diabatic heating is more widely distributed than at 48 h and 60 h shown in Figs. 3f and 4f since convection was more widespread around 54 h. Another perspective of the radial inflow is provided in Fig. 6 that shows horizontal sections of the

radial inflow at 60h at middle and upper levels. At middle levels there is an annulus of strong inflow at a radius of ~40 km. There is some asymmetry with stronger inflow to the north. At upper levels the stronger inflow tends to be to the southeast. Figure 7 shows azimuthally averaged fields at 60 h of the mixing ratio of hydrometeor categories and vapour diffusion (note for brevity the sum of pristine ice crystals and snow, and the sum of graupel and hail are portayed). The sloping base of the stratiform region shown previously in Fig. 4c can be seen to be composed mainly of aggregates and the vapour diffusion is negative, therefore clearly showing that sublimation is taking place. Except near the centre the stratiform base is above the melting level, which strongly indicates that sublimation is playing a major role in causing the mid-level inflow. At low levels near the centre there are also negative values due to evaporation of rain. Aloft in the convective cells there is considerable vapour deposition onto both liquid and ice hydrometeors. Comparing with Fig. 4f the similarity between the fields shows as expected that vapour diffusion is by far the dominant mechanism of latent heat release, with smaller contributions being inferred from freezing and melting.

Returning focus to the earlier time of 48 h Fig. 8 shows horizontal sections of vapour diffusion at several different levels. At the surface the negative values indicate evaporation of rain, which occur in bands and in isolated cells. Significantly there is not widespread drizzle at the surface occurring at this time. Figure 8b at z=4.6 km which is just below the melting level at 4.7 km shows significant negative values occurring in a band in the southeast quadrant. Note that in this panel and the following two panels in this figure the intervals of the colour bars are deliberately chosen to be small in order to show the stratiform component of the diffusion. So the magnitudes in the convective cells are far larger than indicated in the colour bar and appear as red blobs. Figure 8c shows strong sublimation occurring in a ragged annulus at z=6.5 km and a few small patches of vapour deposition which appear to be outside convective cells. Figure 8d at z=8.9 km shows sublimation occurring in a thin ragged outer annulus and also in some small internal patches. On the whole inside the outer annulus of sublimation there tends to be broad stratiform areas of weak vapour deposition.

Figure 9 shows several additional horizontal sections that aid in the interpretation of Fig. 8. The ice field in Fig. 9a shows a band in the southeast quadrant at z=5.5 km, which mainly is composed of aggregates, and it is sublimation below this level that is responsible for the negative vapour diffusion values in Fig. 8b. At this level (z=4.6 km), which is very close to the melting level, the falling aggregates have not had time to melt significantly and convert to rain that can evaporate. The cooling due to sublimation leads to weak subsidence in the band as can be seen in Fig. 9c. Figure 9d shows melting occurring at z=4.3 km, but it is not as large or quite as widespread as the sublimation in the band just above seen in Fig. 8b. The melting produces stratiform rain that can be seen in Fig. 9b, but mixing ratios are small and subsequent evaporation is not going to produce substantial low level cooling.

Convective cells play a fundamental role in the development of a more extensive stratiform cloud canopy aloft during the twelve-hour period from 48-60 h (Figs. 2b and d). As can be seen in Fig. 1 the low-level convective mass flux peaks in the middle of this period and shortly afterwards there is a peak of the total ice content. The strong mid-level inflow that develops by 60 h is clearly to a large degree because of the increased ice content aloft, which can be expected to result in more sublimation. To provide more insight into the role of convective cells we show a cell at 48 h, which was chosen because it is

relatively isolated. The cell is evident in the southern region of the mesoscale band that was discussed in the previous two figures. It is also evident in Fig. 2a and b, at x=25 km, y=-45 km. Figure 10 portrays x/z sections of various fields through the centre of this cell. Figure 10a shows cloud water reaching a height of 10 km, so clearly there are considerable supercooled droplets. Rain is falling out at low levels. Earlier in the cell's lifetime there was also considerable supercooled rain aloft, but at this stage there is mostly ice composed of graupel and hail in the core of the cell, which grew from collection of the rain and cloud water categories at mid-levels, and also by vapour deposition (see Fig. 7f).

The vertical velocity shows a strong updraft aloft and a weak downdraft at low levels. The updraft is very humid (Fig. 10g) indicating there is a significant vertical transport of water vapour taking place. There is also strong diabatic heating in the updraft that is resulting from vapour deposition and from freezing of cloudwater. At upper levels there is an anvil composed of ice that merges with a pre-existing layer of ice produced by previous cells. The downdraft is bringing low values of equivalent potential temperature air to the surface (e.g. Riemer et al 2010 and 2013; Molinari et al. 2013). It is interesting that for the most part the potential temperature perturbation is positive in the downdraft air until it is nearly at the surface, which is probably an effect of water loading. The downdraft air is considerably less humid than the surrounding air at low levels and appears from the wind vectors to be originating from the west side of the cell, although it is difficult to get a complete picture without a three dimensional trajectory analysis. In addition to water loading the downdraft is driven by diabatic cooling as shown in Fig. 10h, which is caused be melting just below the freezing level, and by evaporation of rain. It is evident that the cell is in its late stage of development. The vertical velocity field indicates mid-level convergence, but the complicated flow structure and turbulence surrounding the cell make it difficult to obtain a clear view of how far this convergence extends laterally. There is a possibility that convective cells in the late stage of their development could be contributing to the system scale mid-level inflow, a point that will be returned to in section 4.

Nolan (2007) emphasized that humidity increased due to moist detrainment from deep convective towers prior to a mid-level vortex developing, reaching values exceeding 80 percent below 10km, and this was also found to be the case by NM13. While the Kiljoy et al. (2018) simulation with the ice phase included showed moistening, results were a little different than these previous studies since quite a pronounced humidity peak developed at mid-levels. Fig. 11 shows height versus distance cross sections of the relative humidity through the centre of the domain at 48 h and 60 h. It can be seen that there is generally a significant increase in humidity during the twelve-hour period. At 60 h there are sloping dry notches in the mid-level inflow, which reach to approximately 40 km from the centre. This indicates that the inflow is descending and bringing down drier air from aloft and from the periphery of the system where the air has not been moistened so much by vertical convective moisture transport. NM13 hypothesized the dryness of the inflow air could enhance sublimation and evaporation creating a positive feedback.

At this point we summarize the microphysical processes that are responsible for the three components of stratiform diabatic forcing whose effects on mid-level inflow are investigated in section 4.4. Production of the pristine ice crystal category occurs by heterogeneous nucleation, homogeneous nucleation and splintering, and is particularly large in the cold upper regions of convective towers. They are small crystals that grow by vapour deposition. Larger pristine ice crystals are

categorized as snow, which are able to also grow by riming. These crystals with low fall velocities advect laterally at the cloud tops. By 60 h a significant system scale outflow has developed at a height of 12 km (Fig. 4a) and this is advecting the pristine and snow categories away from the system centre (Fig. 7a). Aggregates are ice particles that form by collision and coalescence of pristine ice, snow, and/or other aggregates. Their fall velocities are faster than pristine ice and snow, and so they tend to accumulate at the lower regions of the stratiform ice layer (Fig. 7b). To a lesser extent this stratiform layer is also composed of graupel that can form from moderate to heavy riming and/or partial melting melting of pristine ice, snow or aggregates (Fig. 7c). The latent heating that occurs in this stratiform ice layer is due to vapour deposition and is relatively weak (Fig. 7f). While Fig. 11 shows that the relative humidity at upper levels and at the periphery of the ice layer is not close to saturation with respect to water, it is supersaturated with respect to ice, otherwise vapour deposition would not be occurring. The effects of this upper-level diabatic heating will be examined in Experiment 3 (Table 1).

As the aggregates and graupel fall to mid levels sublimation occurs and this results in significant cooling at the base of the stratiform ice layer. The effects of this mid-level diabatic cooling will be examined in Experiment 1 (Table 1). Closer to the centre of the system where the humidity is high (Fig. 11), which reduces sublimation, some aggregates and graupel hydrometeors reach lower levels undergoing melting below the freezing level, forming stratiform rain. At this formative stage of the mid-level vortex the stratiform areas at low levels are very patchy (Fig. 8a and b) and the azimuthally averaged diabatic cooling is relatively weak. The effects of this low-level diabatic cooling will be examined in Experiment 2 (Table 1).

## 4.2 The near gradient wind balanced state

Starting from a history file at 45 h a simulation was run for three hours with the method discussed in section 3 for producing a near gradient wind balanced state. Figure 12 shows the modified fields resulting from this procedure that can be compared to the original full physics simulation at 48 h shown in Fig. 3. The tangential velocity shows a more compact maximum aloft than occurs in Fig. 3b, but nevertheless is similar enough that it can be utilized for the purpose of examining the response to imposed diabatic heating and cooling. There is colder air near the surface compared to Fig. 3d, which is due to increased sedimentation velocities during the adjustment period producing higher mixing ratios of rain at low levels, and consequently more evaporative cooling. The central upper level warming has also been reduced during the adjustment. The radial and vertical velocities can be seen to have lessened considerably due to the decay of moist convection.

For completeness Fig. 13 compares the original and modified fields of perturbation pressure and water vapour mixing ratio. The pressure perturbation gradient is less broad aloft consistent with the more compact tangential wind maximum. Vapour mixing ratio has been reduced at low levels because of the low level drying that was applied and the absence of the latent heat flux in the prior three-hour period in spite of the increased low level evaporation that occurred due to the increased sedimentation velocity.

### 4.3 Imposed stratiform forcings

Simple mathematical functions were used to construct diabatic forcings that were reasonable estimates of the three components: mid-level cooling, low-level-cooling, and upper-level warming. Now the diabatic forcings are not constant in time. For instance, it can be seen from Fig. 2 that the upper level canopy becomes more extensive with time, as well as the stronger convective cells becoming more numerous. While the low-level convective mass flux shown in Fig. 1 is a maximum at 55 h, the total ice content peaks a couple of hours later, and presumably the peak stratiform cooling occurs later still as the ice particles fall to lower-levels, which appears to be the case from examining the diabatic heating fields. So instead of imposing a constant forcing over the twelve-hour period it begins weak and then increases linearly with time to the end of the simulation. The forcing functions are initially 30% weaker than the values at the middle of the simulation (6 h simulation time, or 54 h in respect to the full-physics simulation), and 30% stronger by the end of the simulation.

Figure 14 shows azimuthally averaged fields of the diabatic forcing function at 54 h (note that the time will be given in respect to the full physics simulation), and radial velocity, tangential velocity and potential temperature perturbation at the end of the twelve-hour period, for Experiment 1 that has mid-level cooling. The diabatic cooling has maximum amplitude at a radius of 35 km, slopes upward with increasing radius, and falls off sharply below the melting level (z=4.7 km) for radii less than 45 km. At 60 h a mid-level inflow has developed in response to the forcing. Its magnitude is approximately 2.5 m s$^{-1}$ compared to 4.5 m s$^{-1}$ for the full physics simulation (Fig. 4a). It is in the same location, but is notably weaker at a radius of 35 km. An outflow is evident beneath the mid-level inflow. The tangential velocity at middle levels has increased to 11 m s$^{-1}$ and its peak value is in the same location as for the full physics simulation (Fig. 4b). However, it is not as deep and at larger radii the peak values have a more sloping appearance. There is a very cool potential temperature anomaly at the centre between z=4 to 5.7 km, which is slightly above the cool anomaly in Fig. 4d. There is a weak mid-level cool region beyond a radius of 40 km, which has similarities to the full physics simulation. A thin layer of central warming occurs at z=3.5 km, which doesn't have a counterpart in Fig. 4d.

Figure 15 shows results for Experiment 2 that has low-level cooling. The magnitude of the diabatic heating was estimated to be quite weak. Its maximum amplitude is just below the melting level at a radius of 35 km. Note that there is some overlap with the mid-level cooling (Fig. 13a) to prevent sharp cut-offs which are not numerically desirable. This cooling produces a weak inflow at a height of about 5 km, and a weak outflow layer near the surface. There is only a small effect on the tangential winds when compared with the initial state (Fig. 11a). There is a thin layer of cooling at the centre at a height of 4 km and a thin layer of warming immediately above this.

Figure 16 shows results for Experiment 3 that has upper-level warming. The diabatic heating is estimated to be quite weak. The base of the heating slopes upward with distance. This heating drives a weak inflow, which is located just above the inflow produced by the mid-level cooling. This results in a small increase of the tangential winds at a height of 8.5 km. A weak central warm anomaly has formed at a height of 10 km.

Figure 17 shows results for Experiment 4 that sums the three diabatic forcings of Experiments 1-3. The mid-level inflow is increased slightly compared to the mid-level diabatic cooling acting alone. The low-level central cool anomaly is strongest just above 4 km, which agrees with Fig. 4d, but is stronger in magnitude. The combined functions produce a notable central warm anomaly between z=8-10 km, suggesting that the stratiform forcing does contribute significantly to the warm anomaly aloft seen in the full physics simulation.

Figure 18 shows the azimuthally averaged vertical velocity, which has been averaged in time for the last six-hours of Experiment 4 and the azimuthally averaged vapour mixing ratio for the combined functions simulation. Note that the red-blue colour bar for the vertical velocity field does not use a constant interval in order to portray both very weak and very strong values. The vertical velocity is mainly responding to the strong mid-level diabatic cooling. A sloping region of subsidence produced by the diabatic cooling draws in mid-level air above it creating the mid-level inflow. At the centre between z=4-6 km there is very weak upward motion that has a maximum of approximately 1 cm s$^{-1}$. In six hours a parcel of air would be lifted by ~200 m from the 4 km level. Based on the dry adiabatic lapse rate this would result in a potential temperature change of -0.9 K. The potential temperature of the initial sounding increases by ~ 0.9 K between 4.0 km and 4.2 km. So in the latter six hours of the simulation the lifting could result in a cool anomaly of -1.8 K. Therefore it is likely that the main physical mechanism for the cooling in this location is adiabatic ascent, rather than in situ diabatic cooling or lateral mixing.

Comparing the vapour mixing ratio in Fig. 18b with Fig. 13d it can be seen that the subsidence has resulted in considerable drying at mid-levels between r=20-60 km. This would be countered to some extent by vapour sources due to sublimation, and evaporation. A vapour source would also add to the buoyancy, which could influence the descent caused by diabatic cooling and therefore the strength of the mid-level inflow. We ran an experiment adding a water vapour source based on the diabatic cooling rates but did not find a large sensitivity to this factor.

Experiment 5 that examined the impact of precipitation drag by adding a source term to the vertical velocity tendency based on estimated hydrometeor mixing ratios in the stratiform clouds was found to have only a very minor influence on the mid-level inflow. The most clear feature was a very weak upper-level inflow between z=11-12 km and r=40-140 km of 0.1-0.2 m s$^{-1}$ (not shown).

Azimuthally averaged fields of the absolute angular momentum $M = \rho r v + \frac{1}{2}\rho f r^2$ for the full physics simulation are shown in Figure 19 at 48 h and 60 h, as well as the difference field. It is evident that the mid-level inflow has converged angular momentum resulting in a significant increase between r=20-50 km. There has also been a significant increase at low levels due to the inflow caused by convective cells. There appears to be some convective scale vertical transport at a radius of 40 km between z=1-4 km. Convective scale transports are likely responsible for transporting some of the angular momentum converged at mid-levels to the upper troposphere as well, and this is probably why the mid-level tangential wind increase is not as shallow in the full physics simulation compared to the idealized simulation with stratiform mid-level cooling. This issue will be examined in part 2 of this study.

## 4 Conclusions

The results of this study indicate that sublimation of ice at the base of the stratiform ice layer is the major cause of the mid-level inflow occurring in a topical cyclogenesis simulation. Conservation of angular momentum of this inflow air leads to a prominent mid-level vortex. However, the mid-level inflow produced by imposed stratiform forcing is markedly less than occurs in the full physics simulation suggesting that another physical process is also playing an important role. One possibility is that convective cells with a top-heavy heating profile, particularly in the late stage of their evolution, are generating a propagating gravity wave mode having mid-level convergence (e.g. Nicholls et al. 1991; Mapes 1993; Fovell 2002; Adams-Selin and Johnson 2013). Summation of transient gravity waves from multiple cells potentially could cause some system scale mid-level inflow. Results of this study are also suggesting that convective scale momentum transports are taking place that impact the structure of the mid-level vortex. Part two of this study will investigate the possible contribution of diabatic heating in convective cells to supplementing the mid-level inflow and the role of convective scale momentum transports.

An interesting finding is that the low-level central cooling that occurs as the vortex develops is not due to in situ diabatic cooling, but is mainly because of ascent producing adiabatic cooling. This appears to be compensating upward motion in response to diabatically forced sinking in the adjacent air.

The initial formation of the mid-level vortex occurred shortly after a significant peak in the vertical convective mass flux and total ice content. Several oscillations of convective activity occurred during the simulation with a period of 12-15 h (Fig. 1). NM13 hypothesized that these were due to a period of intense convective activity that consumed CAPE, which was then followed by a recovery period with less convective activity until surface fluxes caused CAPE to build up again. NM13 simulations that included radiation showed very pronounced diurnal cycles of convective activity. An interesting subject of future research would be to examine if mid-level vortices tend to develop after a diurnal pulse.

Results of this study contrast with mid latitude studies of mid-tropospheric MCV's in that sublimation cooling of ice at mid-levels appears to be more important than upper level warming and low level cooling in generating a mid-level inflow. It is possible that the greater depth of the troposphere in the tropics may play a role since this might increase the time scale for ice crystals forming aloft to fall below the melting level, thereby giving more time for sublimation to occur in the middle troposphere.

## Appendix A

**Derivation of an equation for latent heating from the changes in the mass of liquid and ice in a grid volume:**

Latent heat released or absorbed in a model time step within a grid volume can be written as the sum of three terms,

$$\delta Q = L_c \delta m_v^l + (L_c + L_f)\delta m_v^i + L_f \delta m_l^i, \tag{A1}$$

where $\delta m_v^l$ is the mass of vapor that condenses to liquid if positive or the mass of liquid that evaporates to vapor if negative, $\delta m_v^i$ is the mass of vapor deposition to ice if positive or the mass of ice sublimated to vapor if negative, and $\delta m_l^i$ is the mass of liquid that freezes to ice if positive or the mass of ice that melts to liquid if negative. Equation (A1) can be rearranged as,

$$\delta Q = L_c(\delta m_v^l + \delta m_v^i) + L_f(\delta m_v^i + \delta m_l^i), \tag{A2}$$

The change of mass within the brackets of the second term is simply $\delta m_i$ the total change in the mass of ice. The mass within the brackets of the first term is the mass of vapor converted to liquid plus the mass of vapor converted to ice, so it is the opposite of the total change in the mass of vapor. Therefore we can write

$$\delta m_v^l + \delta m_v^i = -\delta m_v, \tag{A3}$$

where $\delta m_v$ is the total change in the mass of vapor. Since water mass in a grid volume is conserved prior to sedimentation

then the total change in the mass of vapor is given by,

$$\delta m_v = -(\delta m_l + \delta m_i), \tag{A4}$$

Making these substitutions into Equation (A2) gives,

$$\delta Q = L_c(\delta m_l + \delta m_i) + L_f \delta m_i, \tag{A5}$$

This can be rearranged to give

$$\delta Q = L_c \delta m_l + (L_c + L_f)\delta m_i, \tag{A6}$$

Equation A6 is divided by the mass of dry air in the grid volume to formulate the equation in terms of the heat released or absorbed per unit mass and the mixing ratios of liquid and ice (Eq. 1 of section 3).

*Acknowledgements.* We are extremely grateful to Saurabh Barve for computational assistance, and to Stephen Saleeby who

provided an updated version of the model code which included the binned scheme for cloud droplet riming. We also thank two anonymous reviewers for their constructive comments and suggestions. This research was supported by the National Science Foundation, under Grants NSF AGS 0965721 and NSF AGS 1415244.

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

**Table 1**. List of Experiments

| Experiment | Description |
| --- | --- |
| 1 | Mid-level cooling due to sublimation |
| 2 | Low-level cooling due to melting and evaporation |
| 3 | Upper-level heating due to vapor deposition |
| 4 | Combined diabatic forcing |
| 5 | Precipitation drag |

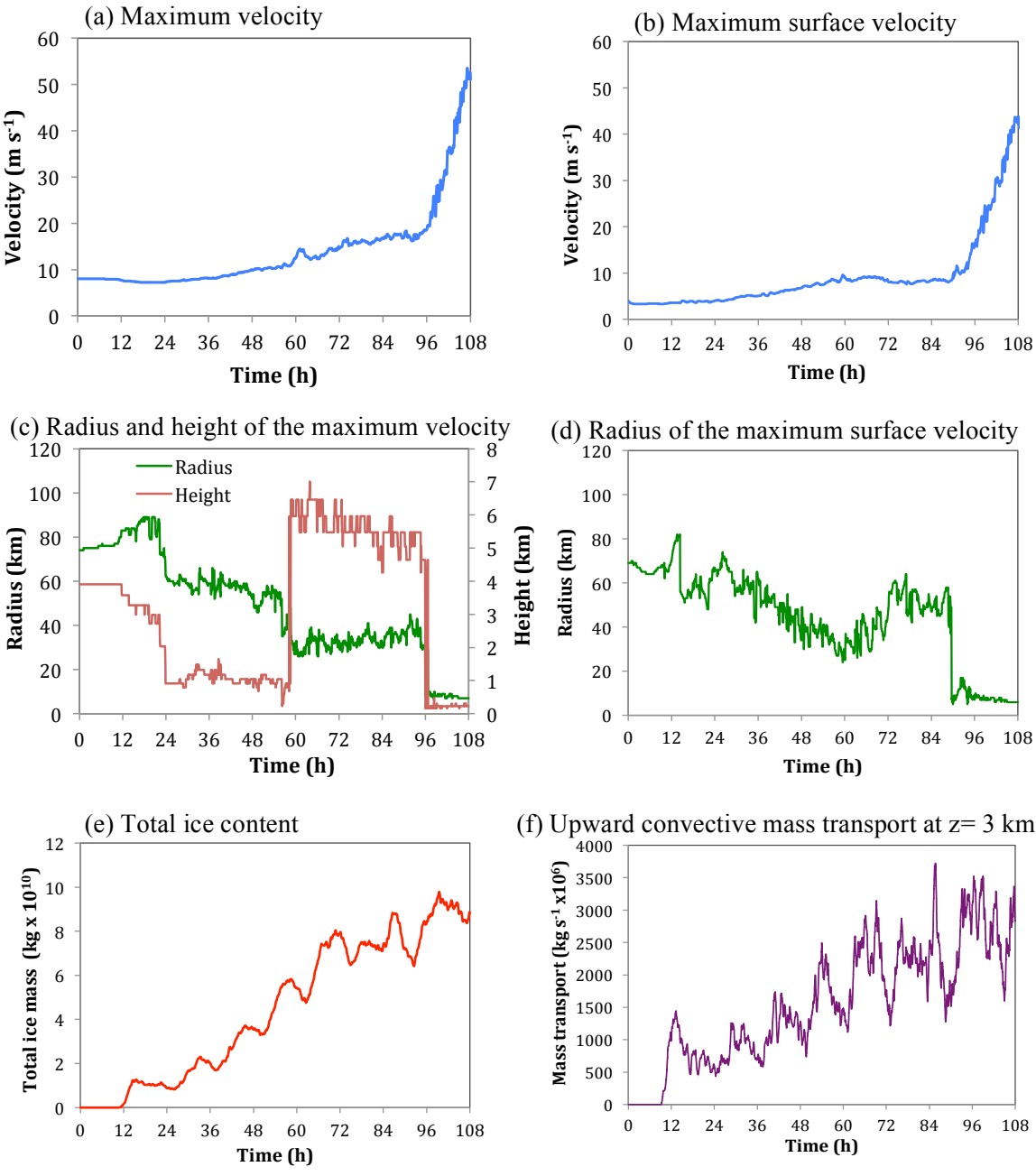

Figure 1. Time series of (a) the maximum azimuthally averaged tangential velocity, (b) the maximum azimuthally averaged surface tangential velocity, (c) the radius and height of the maximum azimuthally averaged tangential velocity, (d) the radius of the maximum azimuthally averaged surface tangential velocity, (e) the total ice mass content in the fine grid domain, and (f) the total upward mass transport by updrafts greater than 0.5 m s[-1], within a radius of 100km from the center of the domain, at a height of 3 km.

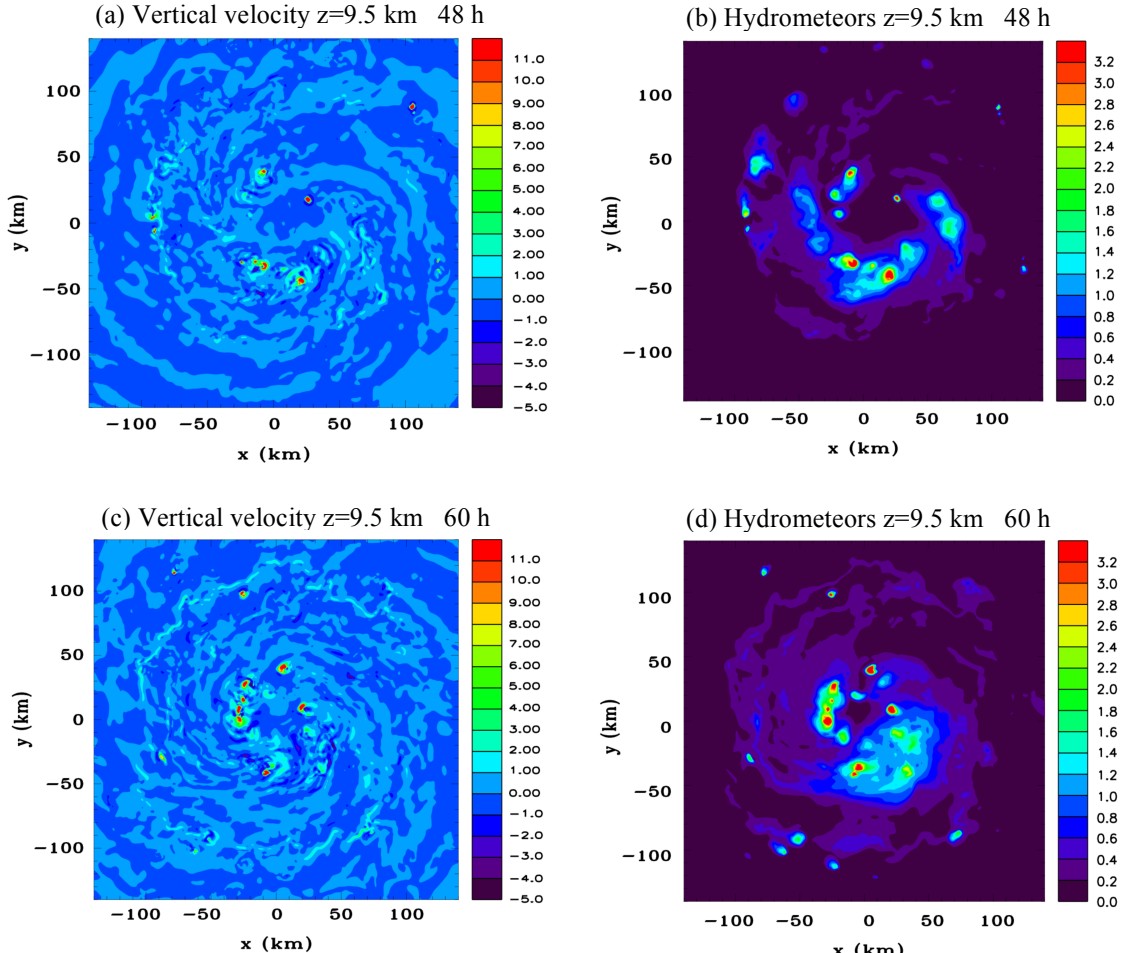

Figure 2. Horizontal sections of vertical velocity (m s[-1]) and total hydrometeor mixing ratio (g kg[-1]) at 48h (**a**) and (**b**) respectively, and at 60 h (**c**) and (**d**) respectively, at z=9.5 km.

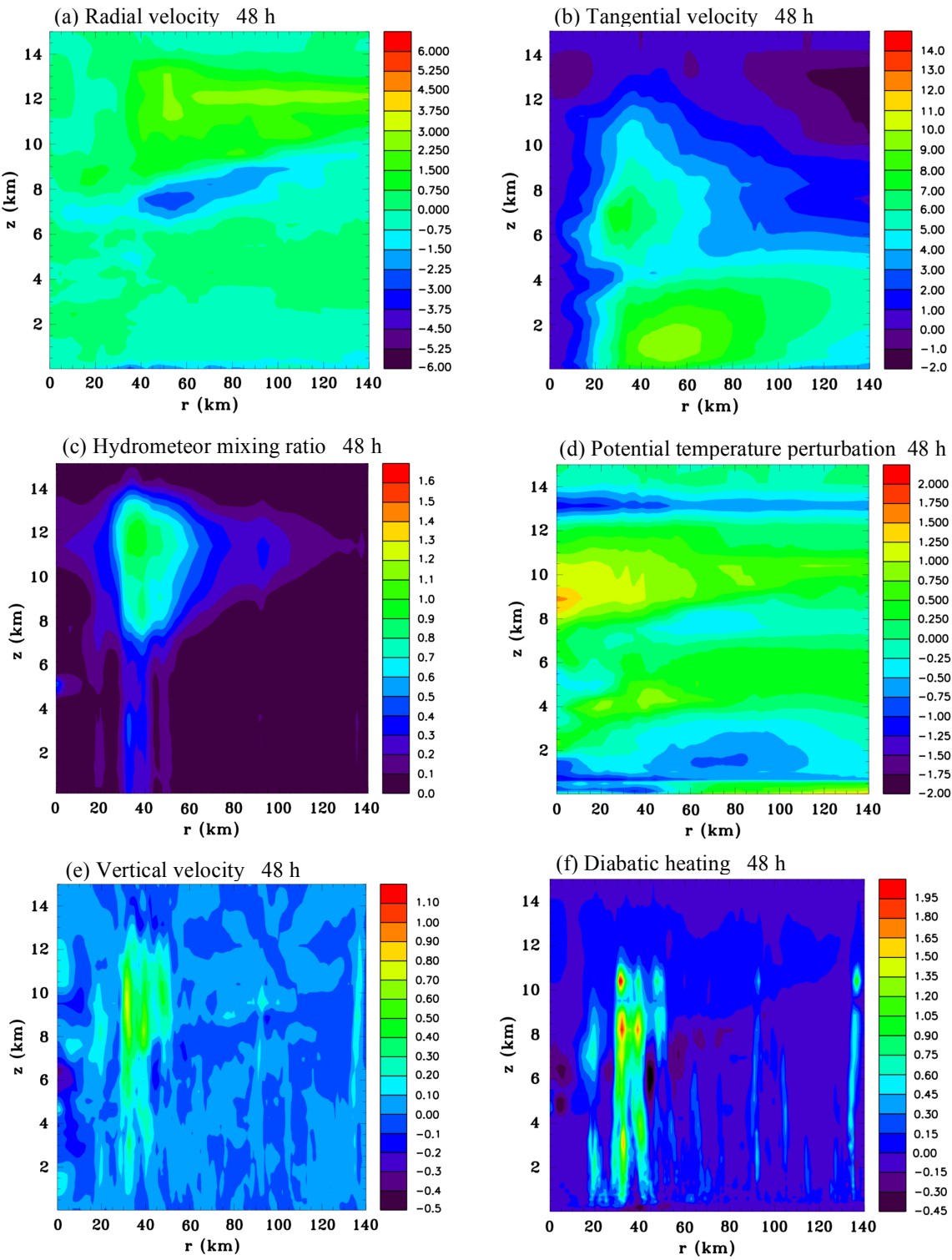

Figure 3. Azimuthally averaged vertical sections of (a) radial velocity (m s[-1]), (b) tangential velocity (m s[-1]), (c) hydrometeor mixing ratio (g kg[-1]), (d) perturbation potential temperature (K), (e) vertical velocity (m s[-1]), and (f) diabatic heating rate (J kg[-1] s[-1]), at 48 h.

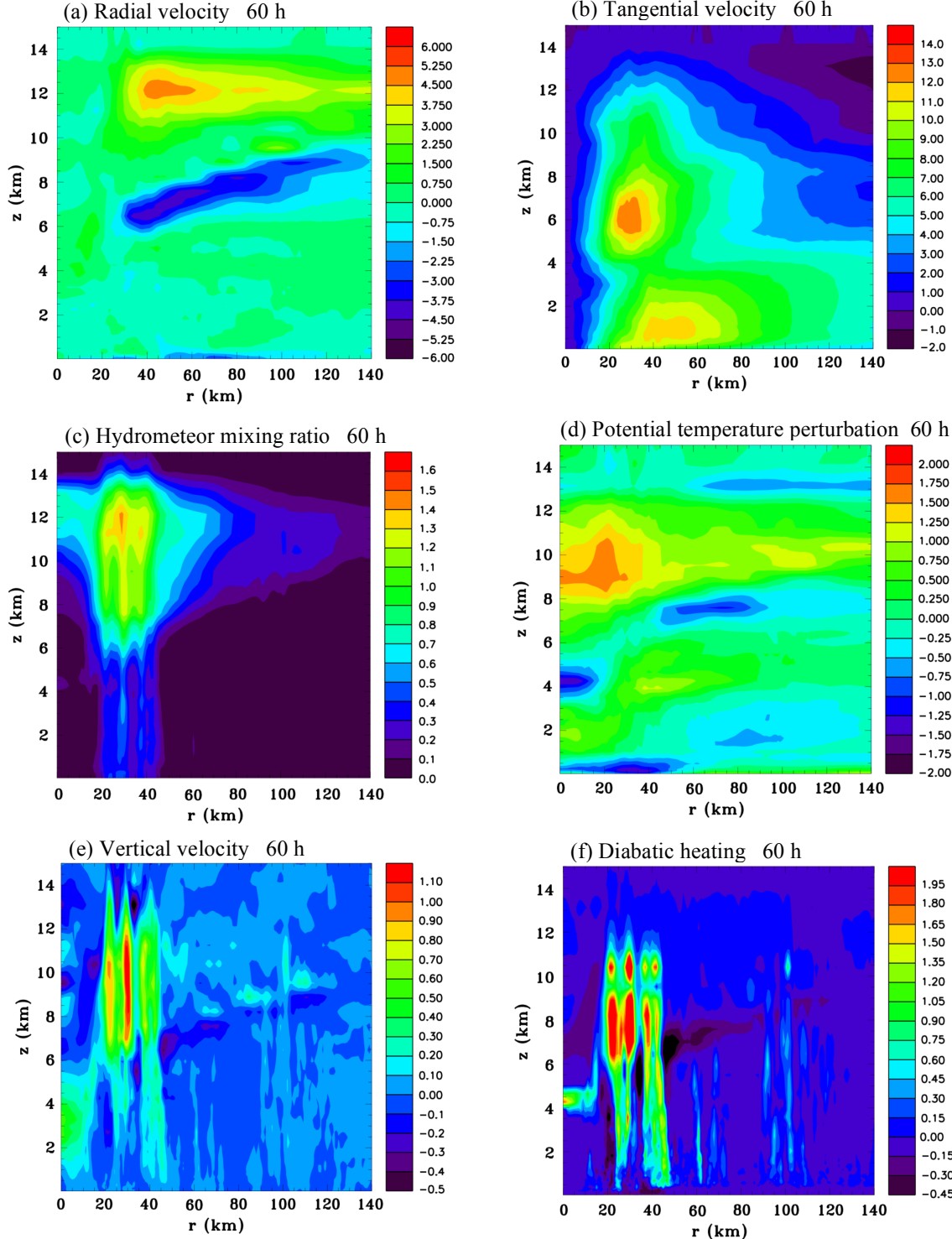

Figure 4. As in Figure 3, but at 60 h.

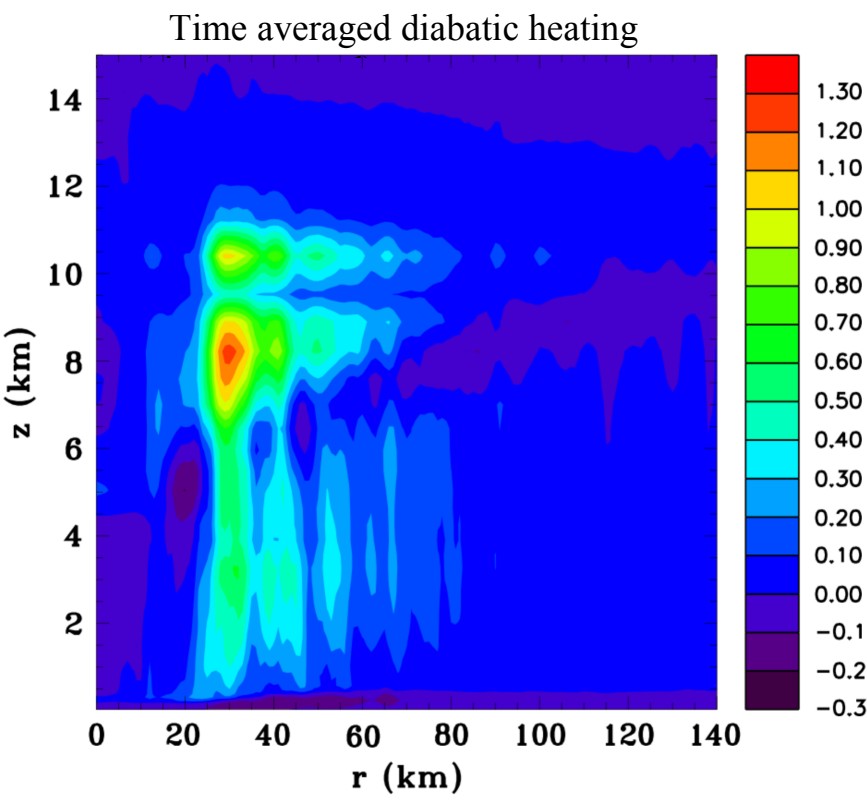

Figure 5. Time and azimuthally averaged vertical section of diabatic heating rate (J kg$^{-1}$ s$^{-1}$), between 48h to 60h.

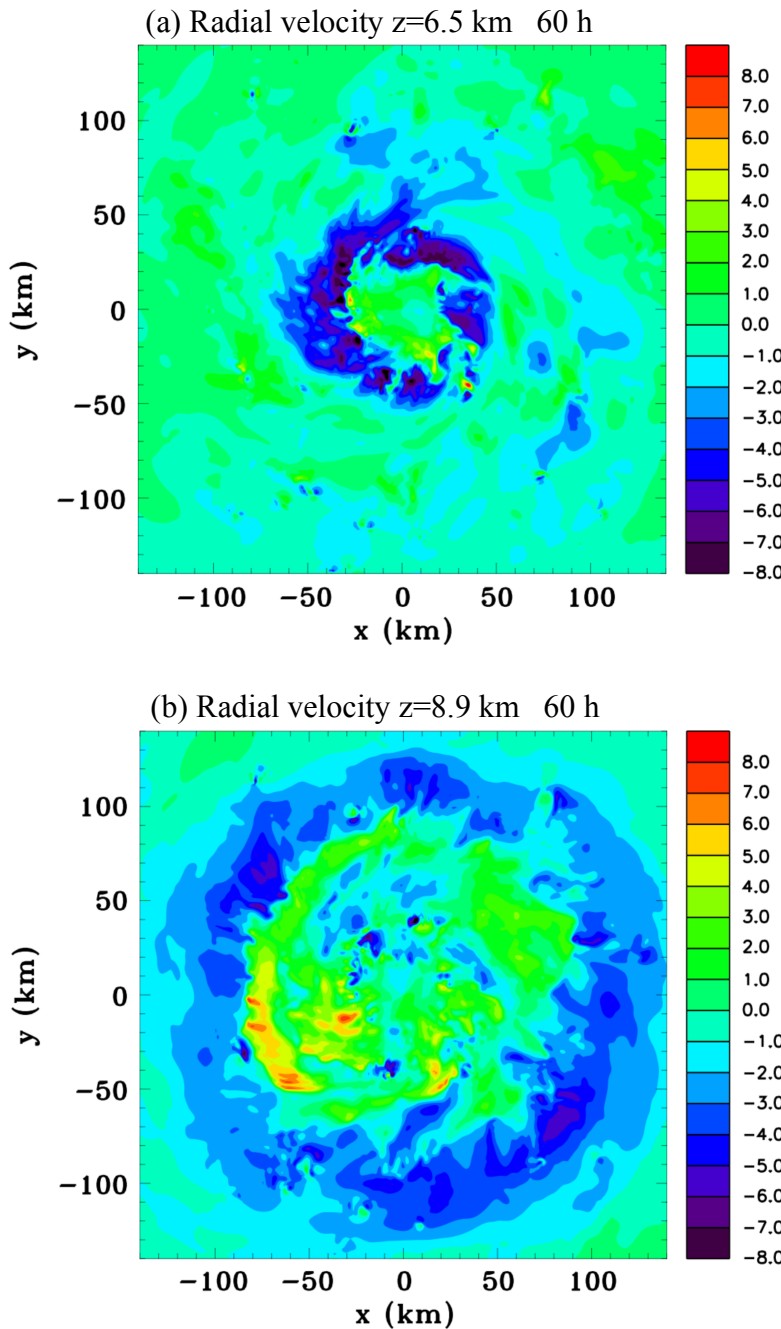

Figure 6. Horizontal sections of radial velocity (m s$^{-1}$), at 60 h:
(**a**) z=6.5 km (**b**) z=8.9 km.

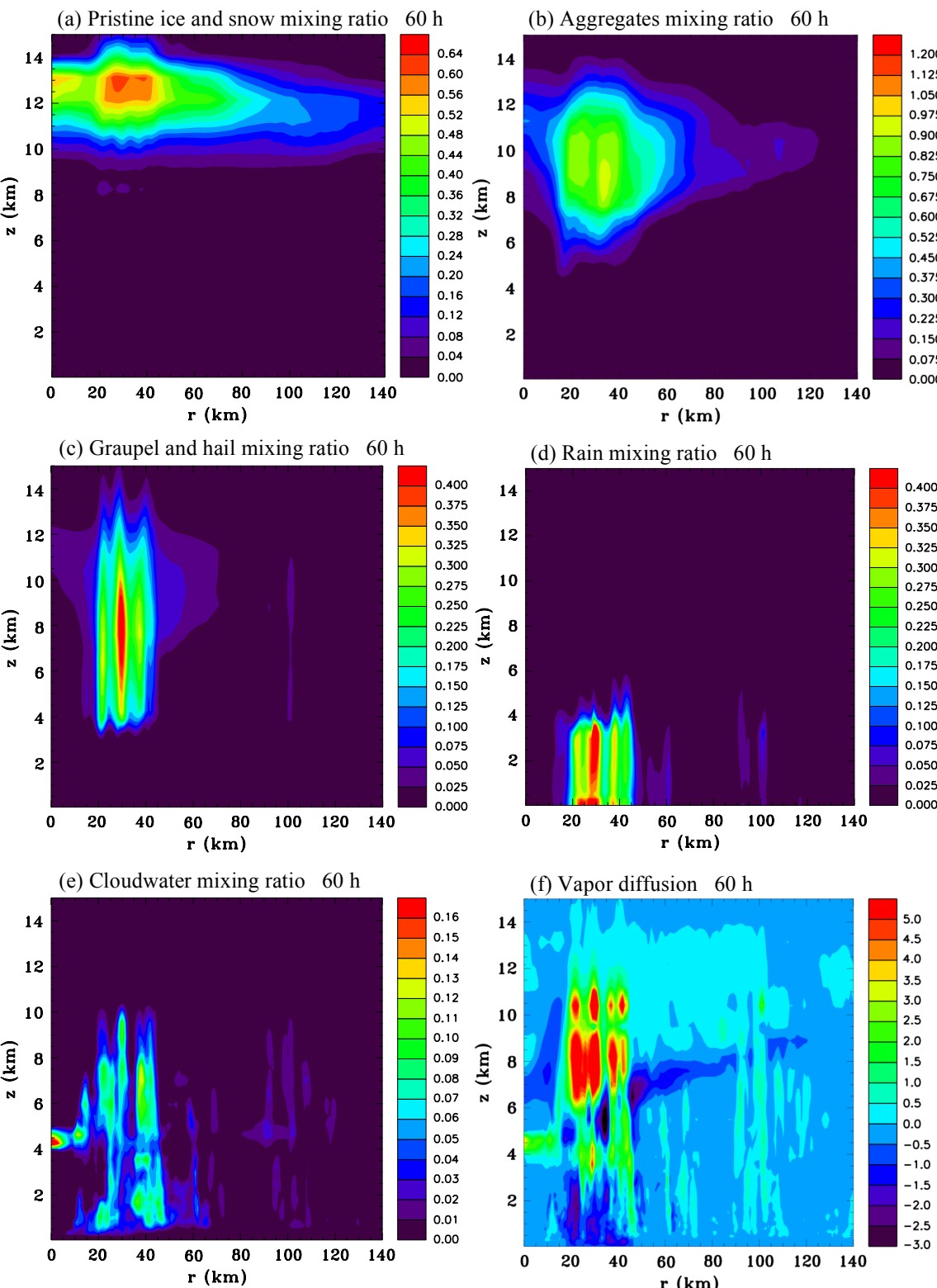

Figure 7. Azimuthally averaged vertical sections of hydrometeors and vapor diffusion at 60 h: Mixing ratios (g kg$^{-1}$) of (**a**) the sum of pristine ice and snow, (**b**) aggregates, (**c**) the sum of graupel and hail, (**d**) rain, and (**e**) cloudwater. (**f**) The rate of change of the sum of liquid and ice mixing ratios due to vapor diffusion ($\times 10^{-7}$ kg kg$^{-1}$ s$^{-1}$).

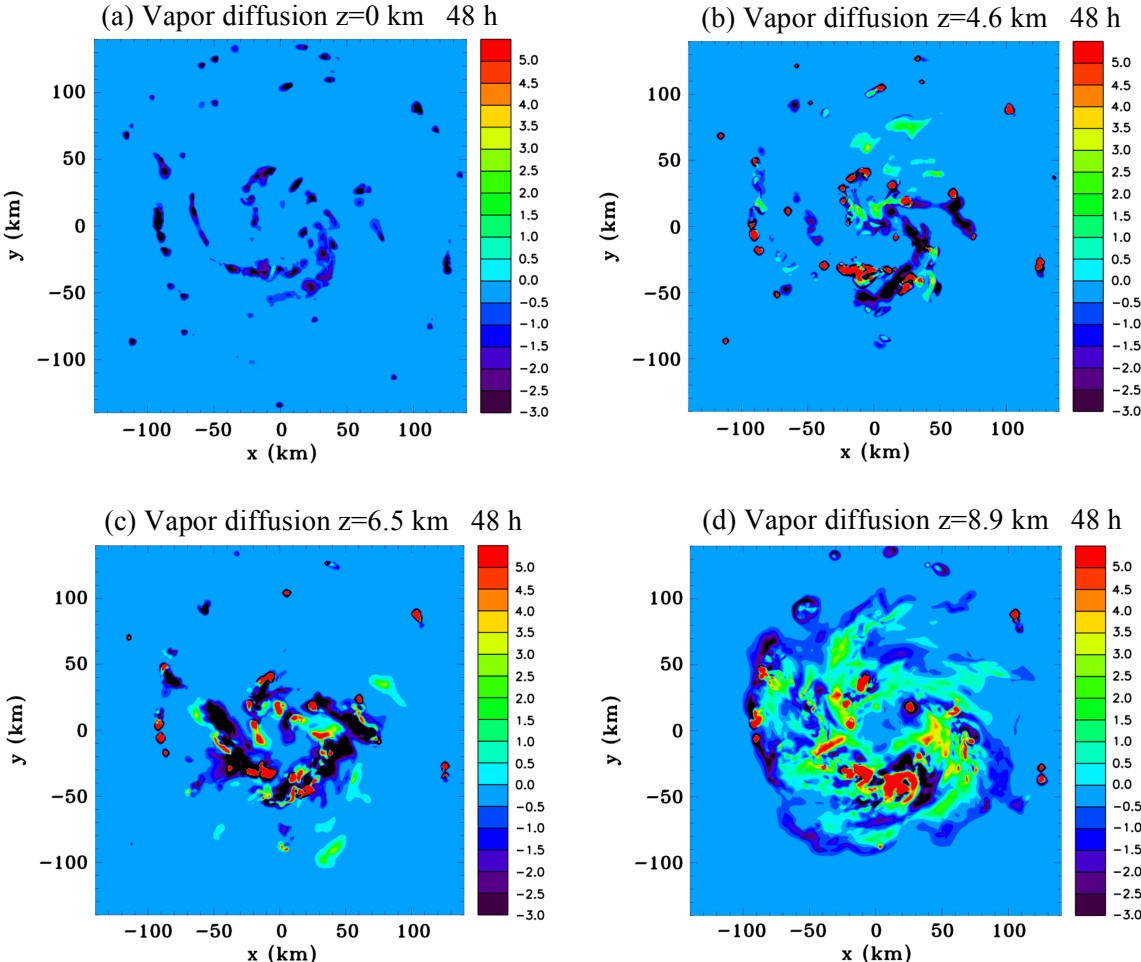

Figure 8. Horizontal sections of the rate of change of the sum of liquid and ice mixing ratios due to vapor diffusion ($\times 10^{-7}$ kg kg$^{-1}$ s$^{-1}$), at 48 h: (**a**) z=0 km, (**b**) z=4.6 km, (**c**) z=6.5 km, and (**d**) z=8.9 km.

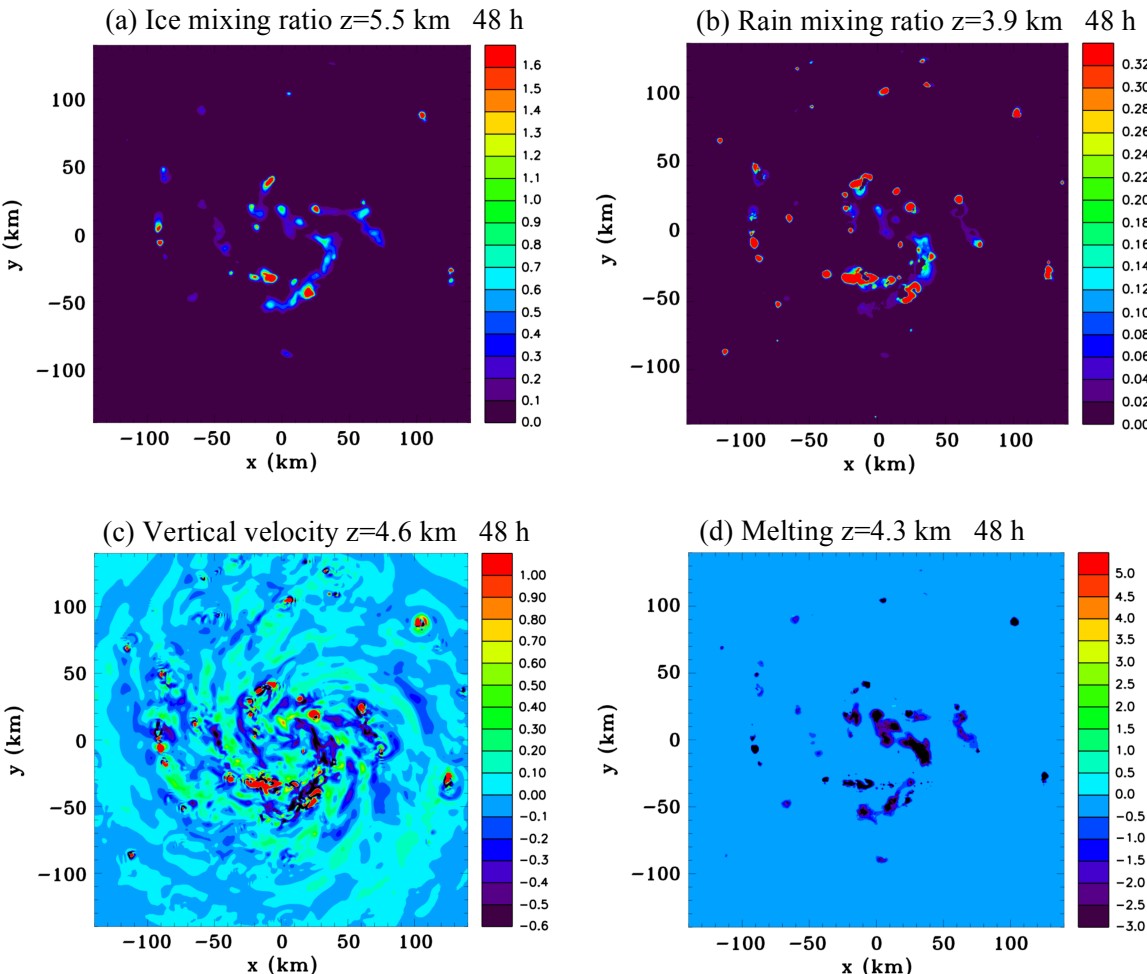

Figure 9. Horizontal sections at 48 h of: (**a**) ice mixing ratio (g kg[-1]) at z=5.5 km, (**b**) rain mixing ratio (g kg[-1]) at z=3.9 km, (**c**) vertical velocity (m s[-1]) at z=4.6 km, and (**d**) rate of change of ice mixing ratio due to melting (×10[-7] kg kg[-1] s[-1]) at z=4.3 km.

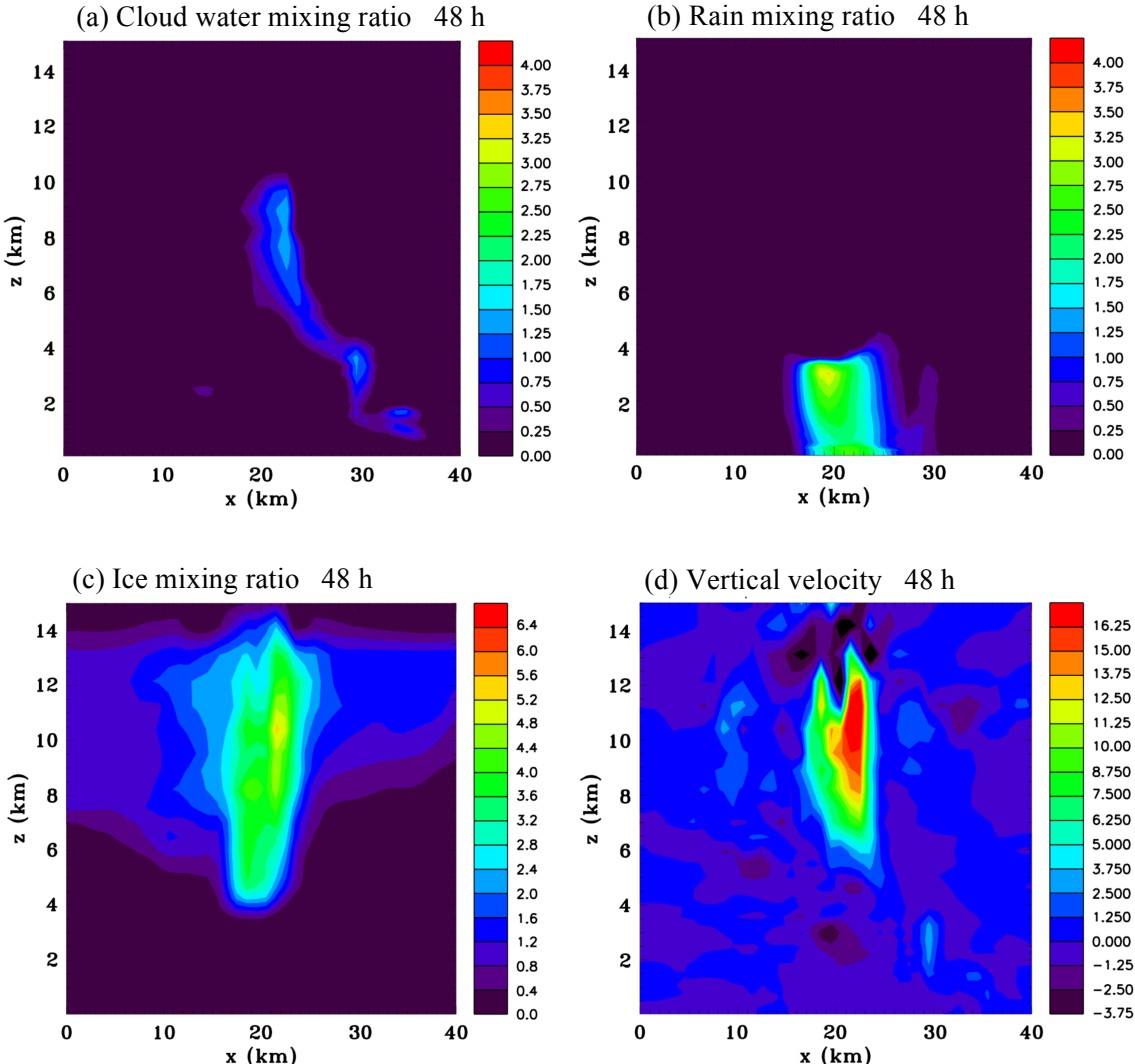

Figure 10. Vertical sections through a convective cell at x=25 km, y=-45 km, at 48 h, of (**a**) cloud water mixing ratio (g kg$^{-1}$), (**b**) rain mixing ratio (g kg$^{-1}$), (**c**) ice mixing ratio (g kg$^{-1}$), (**d**) vertical velocity (m s$^{-1}$), (**e**) equivalent potential temperature (K), (**f**) potential temperature perturbation (K), (**g**) Relative humidity (RH) and wind vectors (maximum 15 m s$^{-1}$), and (**h**) diabatic heating rate (J kg$^{-1}$ s$^{-1}$)

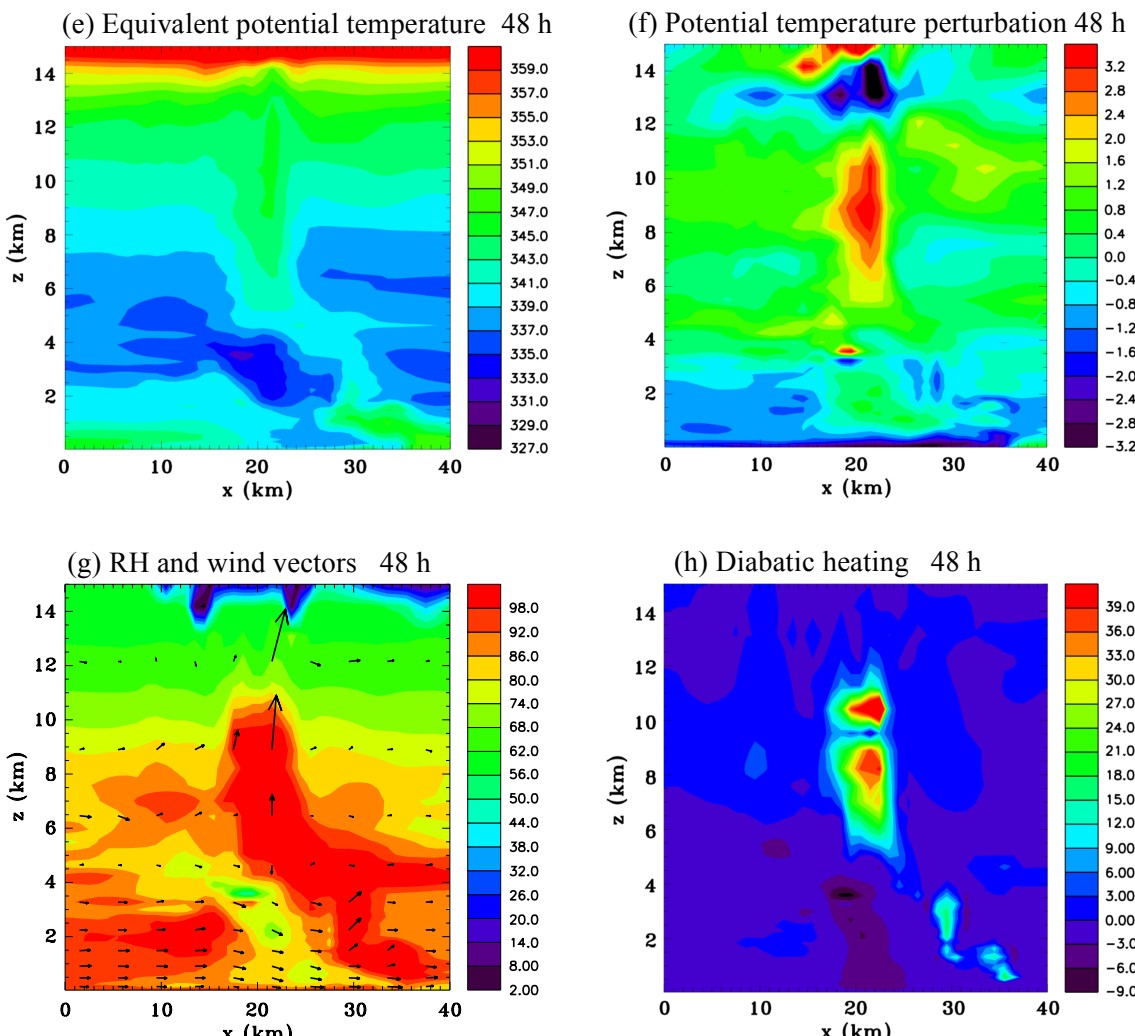

Figure 10. Continued

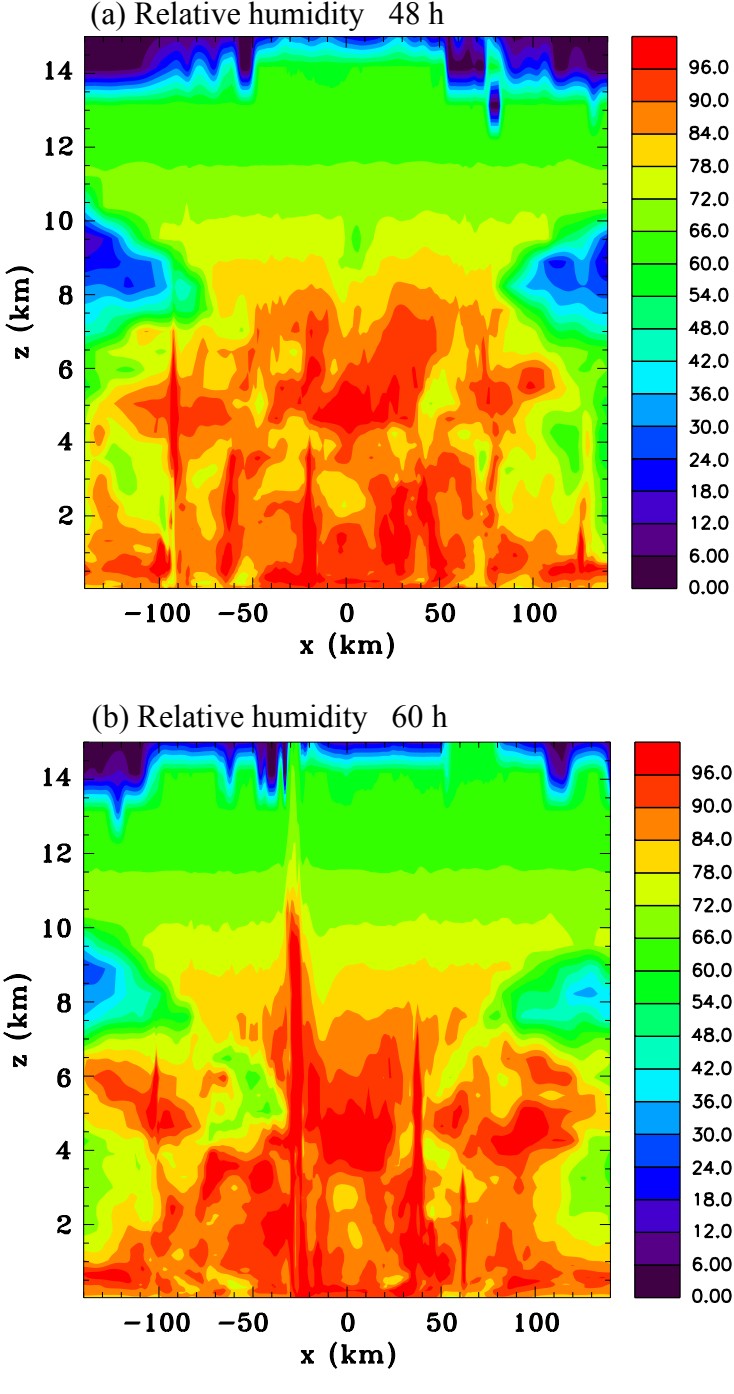

Figure 11. Vertical sections of relative humidity with respect to water (percent) through the centre of the domain: (**a**) 48 h (**b**) 60 h.

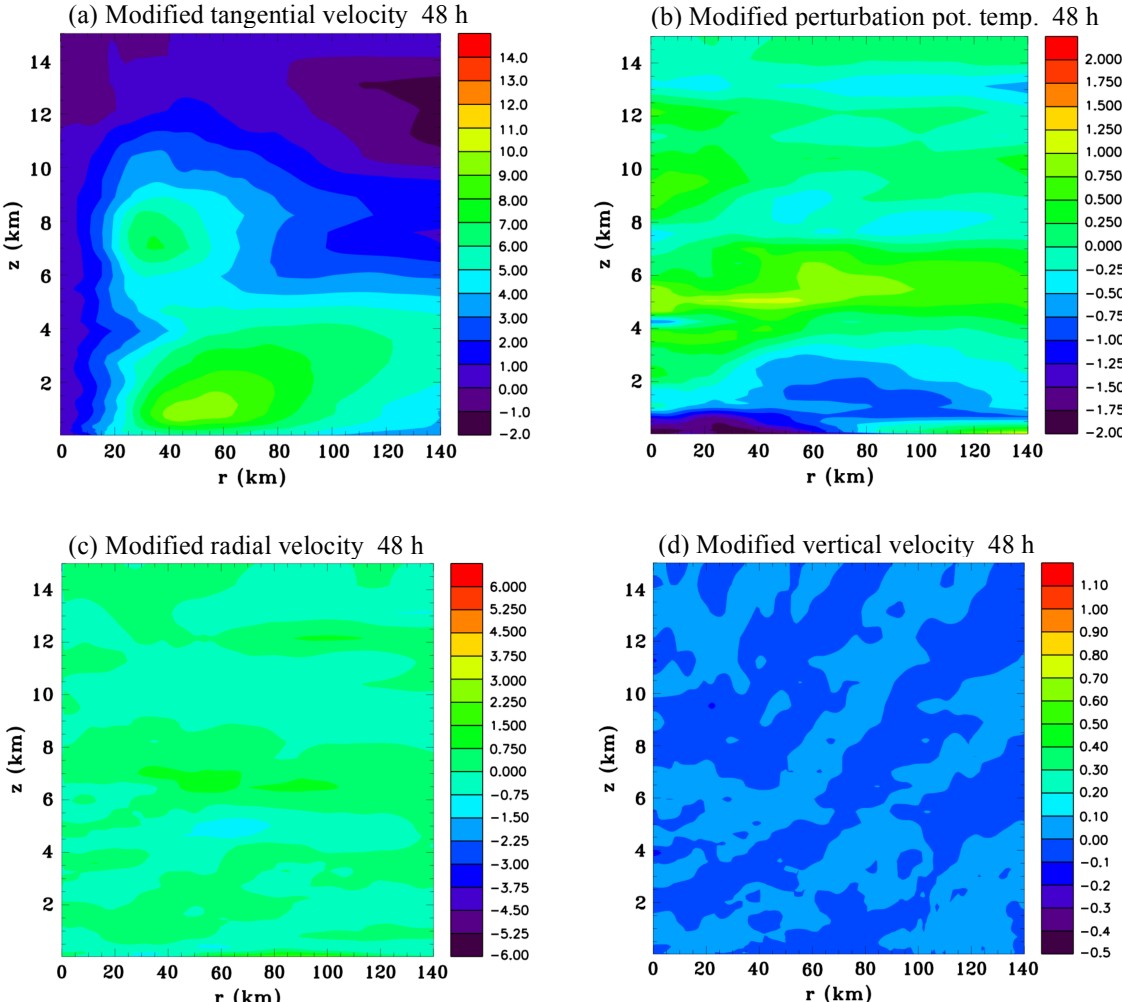

Figure 12. Modified near gradient wind balanced azimuthally averaged fields of (**a**) tangential velocity (m s$^{-1}$), (**b**) perturbation potential temperature (K), (**c**) radial velocity m s$^{-1}$), and (**d**) vertical velocity (m s$^{-1}$), at 48 h.

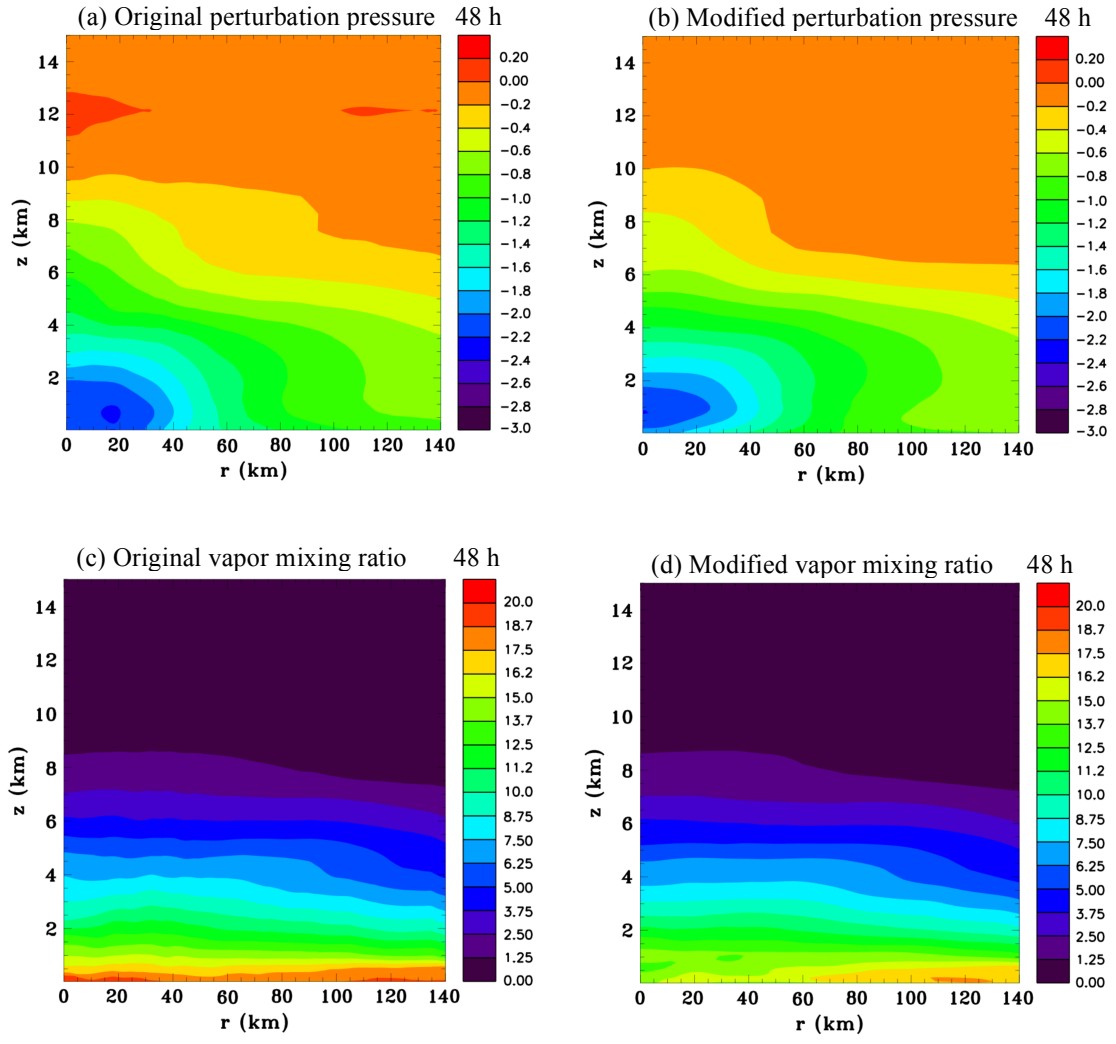

Figure 13. Original and modified near gradient wind balanced azimuthally averaged fields of pressure perturbation (mb), (**a**) and (**b**) respectively, and vapor mixing ratio (g kg$^{-1}$), (**c**) and (**d**) respectively, at 48 h.

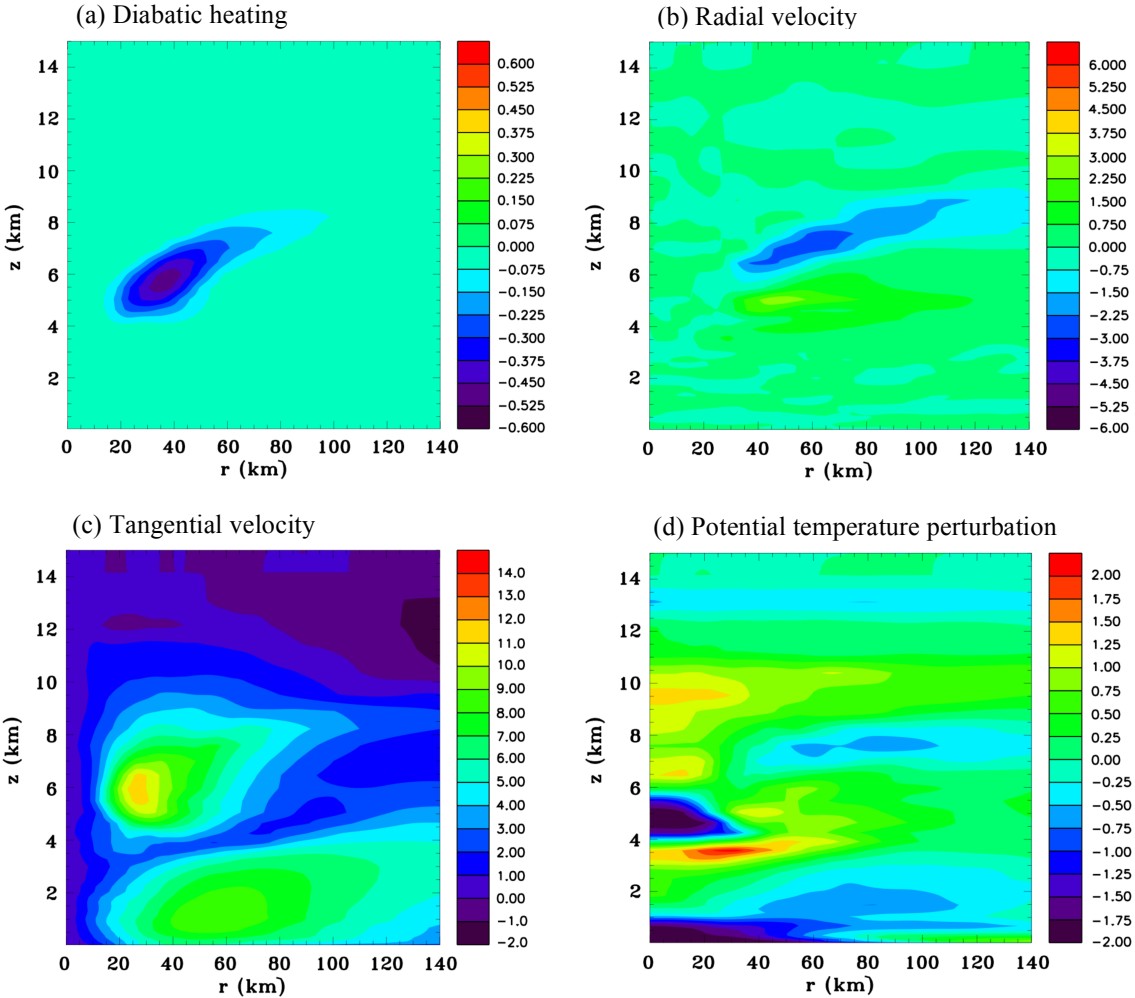

Figure 14. Azimuthally averaged vertical sections at 60 h for the mid-level cooling function. (**a**) Diabatic heating rate (J kg$^{-1}$ s$^{-1}$), (**b**) radial velocity (m s$^{-1}$), (**c**) tangential velocity (m s$^{-1}$), and (**d**) perturbation potential temperature (K).

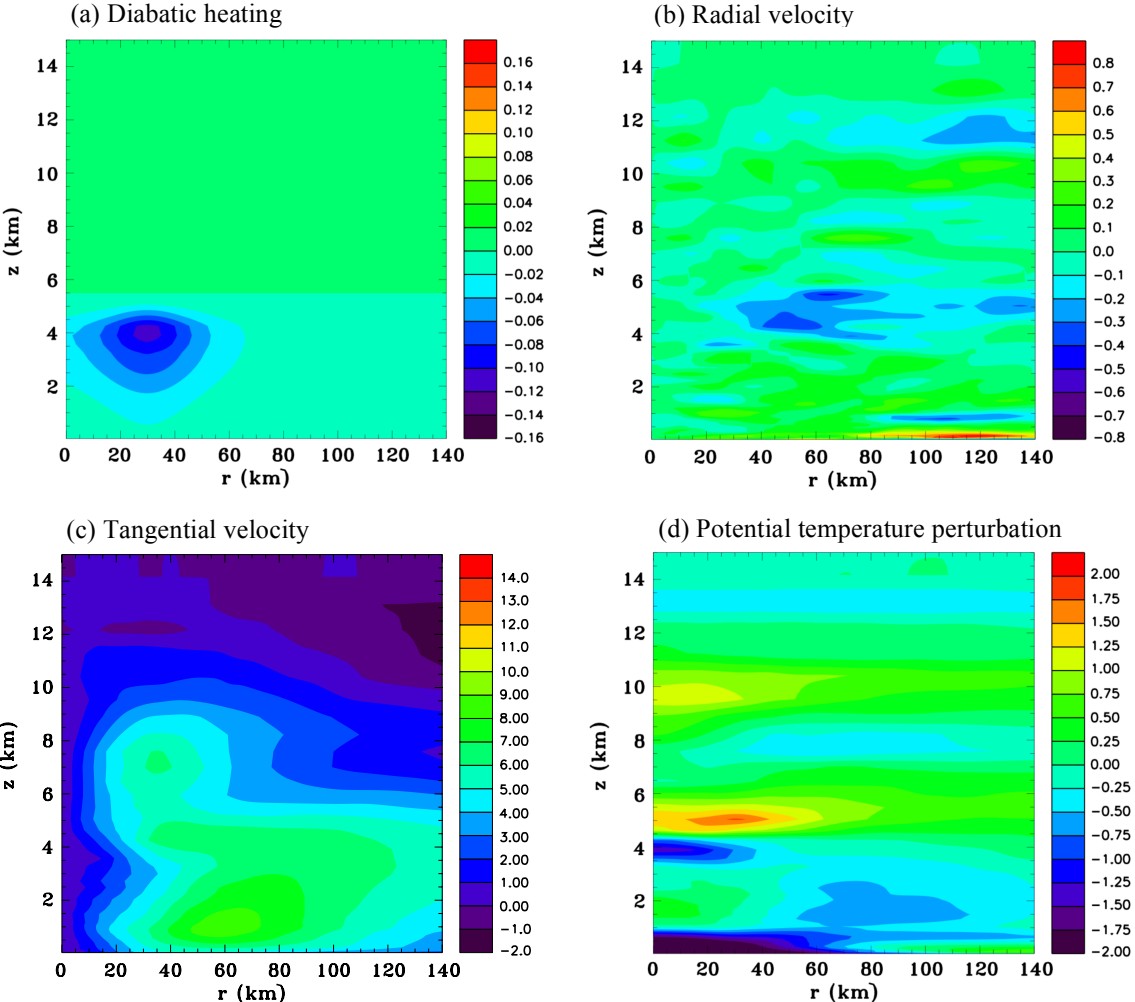

Figure 15. Azimuthally averaged vertical sections at 60 h for the low-level cooling function. (**a**) Diabatic heating rate (J kg$^{-1}$ s$^{-1}$), (**b**) radial velocity (m s$^{-1}$), (**c**) tangential velocity (m s$^{-1}$), and (**d**) perturbation potential temperature (K).

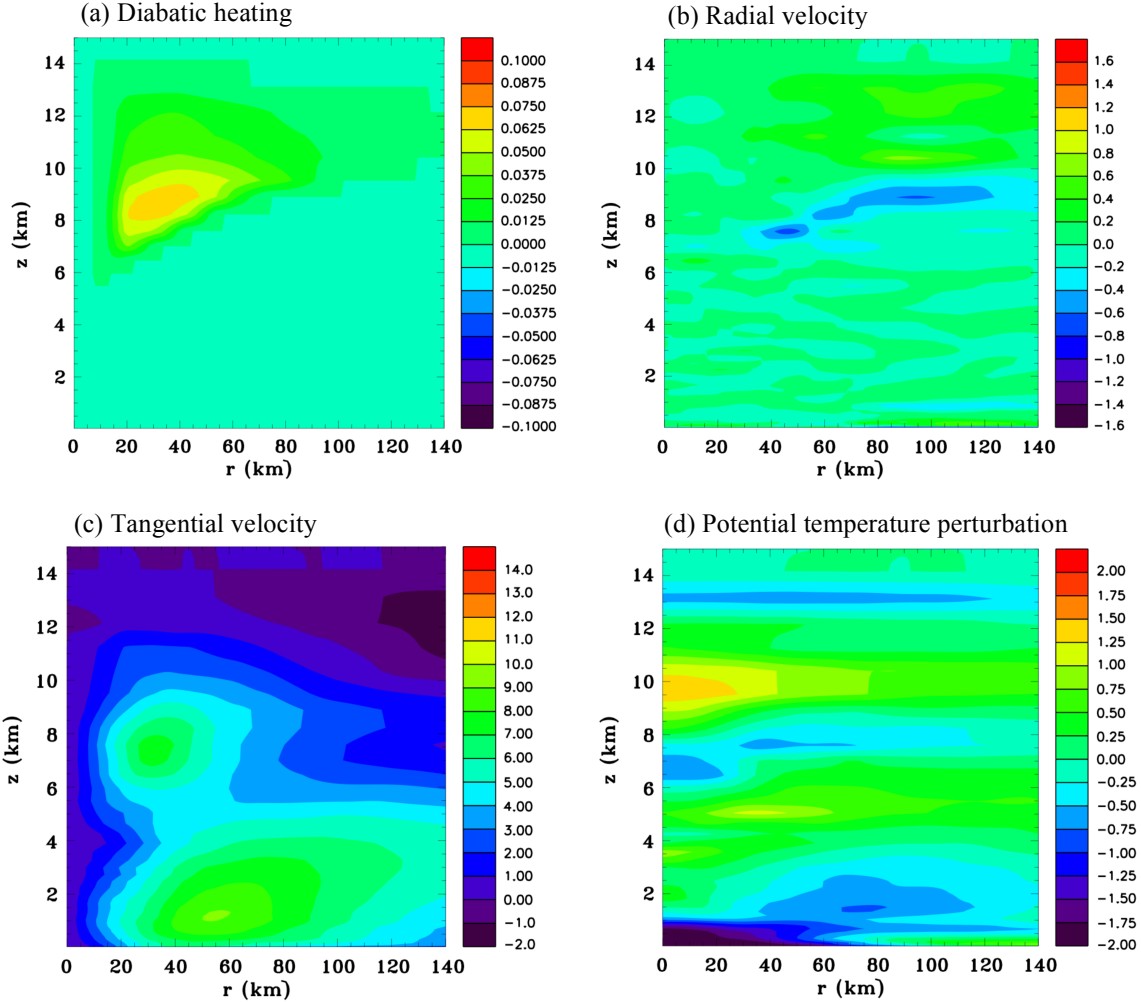

Figure 16. Azimuthally averaged vertical sections at 60 h for the upper-level heating function. (**a**) Diabatic heating rate (J kg$^{-1}$ s$^{-1}$), (**b**) radial velocity (m s$^{-1}$), (**c**) tangential velocity (m s$^{-1}$), and (**d**) perturbation potential temperature (K).

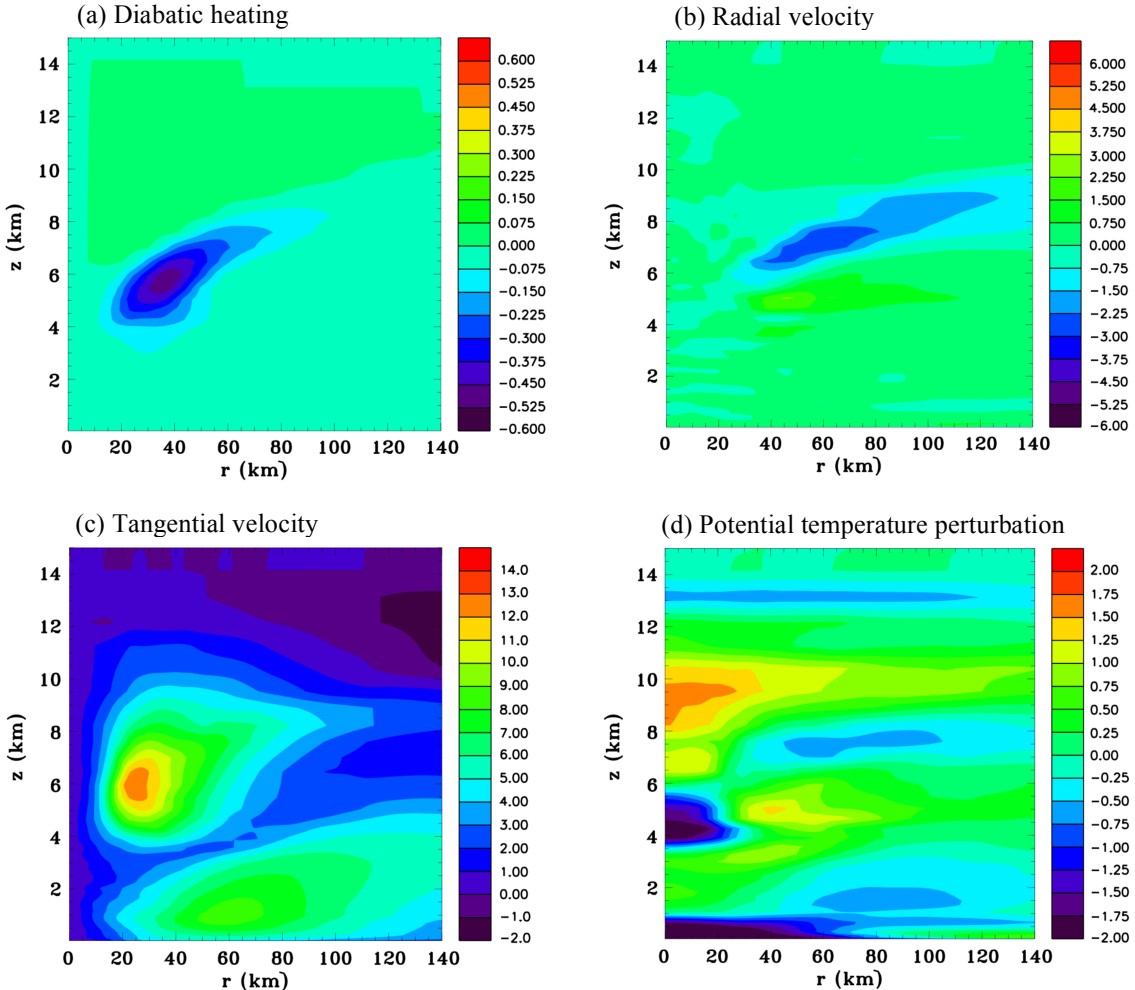

Figure 17. Azimuthally averaged vertical sections at 60 h for the combined diabatic forcing functions. (**a**) Diabatic heating rate (J kg$^{-1}$ s$^{-1}$), (**b**) radial velocity (m s$^{-1}$), (**c**) tangential velocity (m s$^{-1}$), and (**d**) perturbation potential temperature (K).

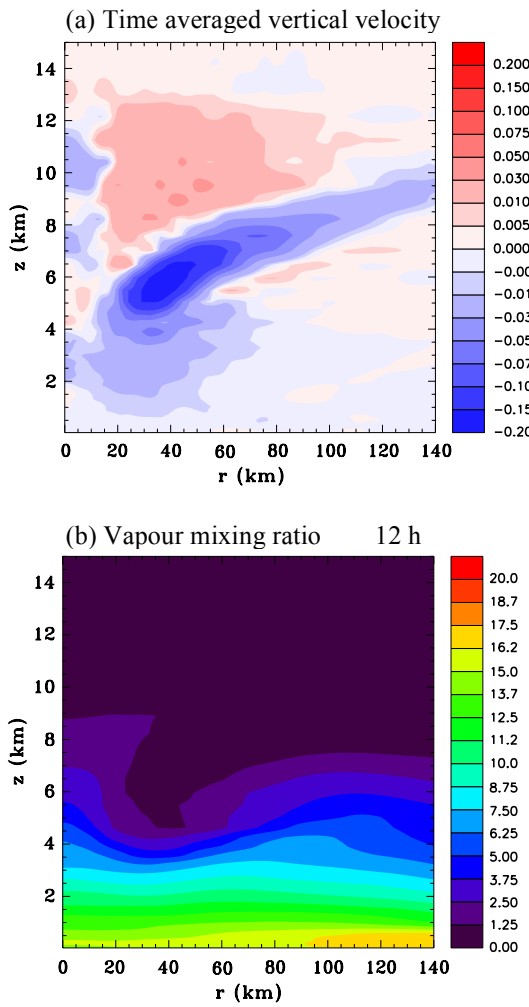

Figure 18. Azimuthally averaged vertical sections for the combined diabatic forcing functions simulation. (**a**) Time average of vertical velocity between 54-60 h (m s$^{-1}$), and (**b**) vapor mixing ratio at 60 h (g kg$^{-1}$).

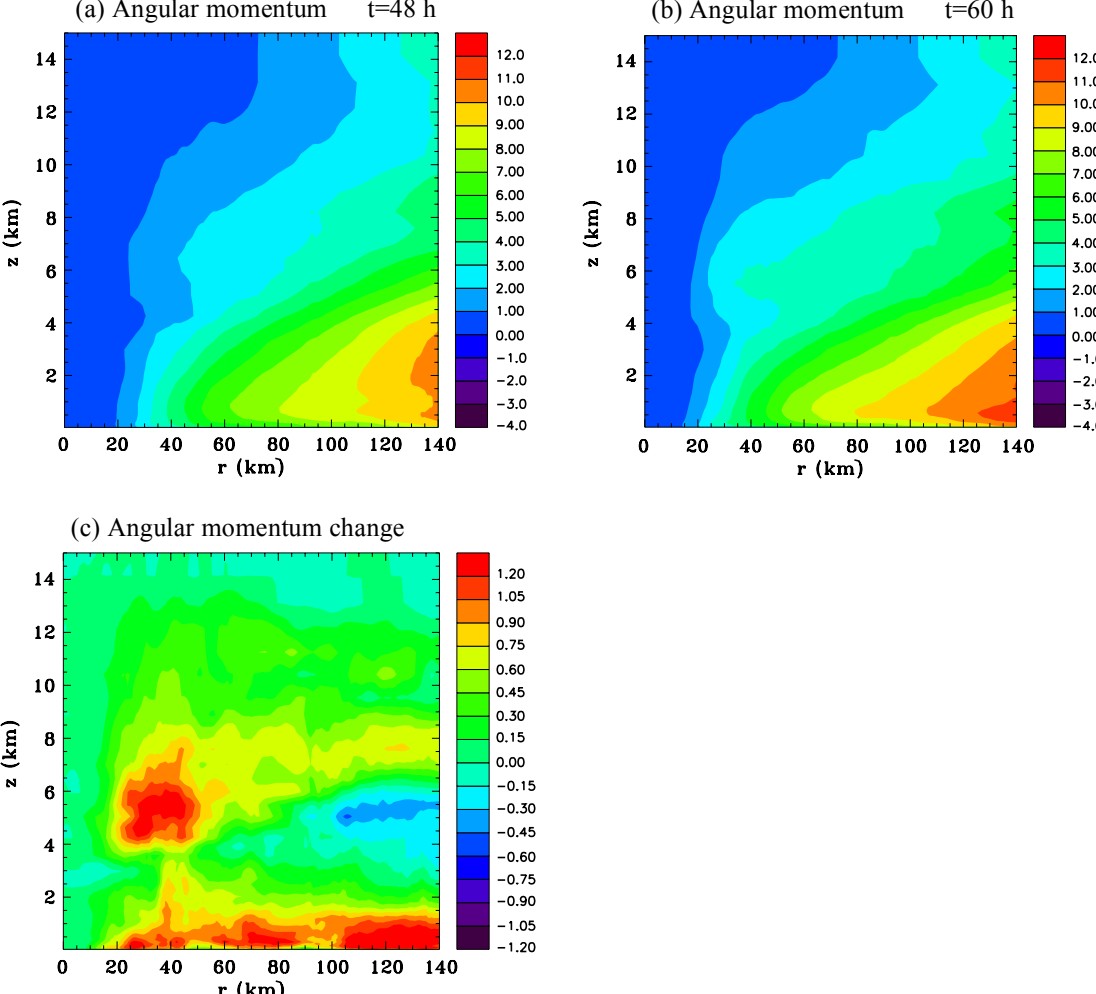

Figure 19. Azimuthally averaged fields of angular momentum. (**a**) At t=48 h, (**b**) t=60 h, and (**c**) the difference field, (kg m$^{-1}$ s$^{-1}$), for the full physics simulation.