# Peer review of "A numerical modeling investigation of the role of diabatic heating and cooling in the development of a mid-level vortex prior to tropical cyclogenesis. Part I: The response to stratiform components of diabatic forcing"

_Atmospheric Chemistry and Physics, 2018_

## Referee Comment (RC1) · Anonymous Referee #1 · 30 May 2018

Review of

A numerical modeling investigation of the role of diabatic heating and cooling in the development of a mid-level vortex prior to tropical cyclogenesis. Part I: The response to stratiform components of diabatic forcing

Ref.: acp-2018-316

**Recommendation:** Minor revision

The authors investigate the reasons why a mid-level vortex develops during tropical cyclogenesis by isolating the effects of the stratiform components of heating and cooling. They find that the cooling from sublimation at mid-levels is primarily responsible for the bulk of the inflow at this height. As air parcels moving inwards at mid-levels conserve their absolute angular momentum, a mid-level vortex subsequently develops. This is a very useful study that sheds light on why mid-level vortices form in some numerical models, although there is some debate as to whether mid-level vortices are necessary for genesis. I recommend that the paper be accepted with minor comments and I have some specific questions/recommendations for the authors.

**General comments:**

The article is well written and presented, although I would request more discussion and description in the results section. Most of my comments are relatively minor.

1. The method employed by the authors to isolate the effects of the diabatic heating/cooling profiles seems particularly messy. I wonder whether it is the "best approach". The authors suggest that the vortex will be "not too different from the quasi-balanced state ..." although there appears to me to be lots of arbitrary decisions made to achieve this state. Is it not simpler and cleaner to take the averaged diabatic heating profiles and run the Sawyer-Eliassen model with a balanced vortex?

2. In the introduction (page 3, lines 7-14) the authors describe a theory where a mid-level vortex descends to the surface. A vortex descending to the surface would violate Haynes and McIntyre (1987). There can be no net downward transport of vorticity from the middle level vortex, ruling out the possibility of genesis being a ``top down" process. In any case, down flow from mid-levels would produce outflow near the surface, leading to a dilution of the vertical vorticity transported downward.

3. Also in the introduction (page 3, lines 20-24) the authors introduce the theory that the mid-level vortex is conducive to convection with a more bottom heavy mass profile. A recent paper by Kilroy *etal*. 2018 found the opposite: namely, a simulation with warm-rain-only microphysics did not develop a mid-level vortex, but the convection had a bottom heavy mass flux profile. The simulation with ice-microphysics developed a mid-level vortex, but did not have a bottom heavy mass flux profile (see their Fig. 13).

4. Page 4, lines 17-19: "Overall we consider that there is **ample evidence** that mid-level vortices **may at times** be playing an active role in tropical cyclogenesis, and even though they are **unlikely to always be essential** to the process, there are going to be **consequences** when a mid-level vortex develops which need to be further investigated" This sentence is very vague and confusing. How can there by ample evidence that it may at times play an active role? What is the active role a mid-level vortex plays? What are the consequences? Unlikely to

always be essential, is there any evidence they are ever essential? I won't argue that mid-level vortices occur in numerical models (and in nature), but is their formation a consequence of ice microphysics (or other model parameters) and not a necessity for genesis?

5. On that note, have the authors considered using/redoing one of their simulations from NM13 which followed Pathway 1 (and included ice-microphysics) and doing a similar analysis? It would be interesting to understand why a similar simulation with ice did not develop a mid-level vortex. Surely these simulations would also contain ample mid-level cooling from sublimation.

6. Start of section 4. I found the results section a little difficult to follow at first, as there was little to no description of the vortex evolution. I have no feel for vortex strength or development. There should be a better segue into the main results. Perhaps the authors could think about including a plot showing the time evolution of the maximum tangential winds, radial winds, etc. and give a description of the vortex evolution. Another section that would be greatly improved by proper introduction is from line 19, page 10. Why the jump from a system analysis to describing a single cell in detail. Why is this cell in particular important? I wasn't sure of the significance of these results.

**Minor comments:**

- Page 3, line 4. "cod pool"

- Page 4, lines 12-13. "Also Hurricane Guillermo (1991) in the Eastern Pacific by Bister and Emanuel…". Perhaps you can rephrase this, it reads like Bister and Emanuel were responsible for the hurricane.

- Page 7, line 13. "conservative" should be "conserved".

- Page 8, line 31 and page 9, line 14: I dislike the phrase "descending inflow". The plot is not showing any vertical motion. Do you mean that the height of the inflow is decreasing with decreasing radius?
- Page 9, line 1: "This mid-level maximum of tangential winds appears to be associated with the midlevel inflow". Why "appears" to be, can there be any other reason?
- Page 9, lines 4-5. Can this low level cooling be from the initial vortex, even after 48 h has passed?
- Page 9, line 6. Do you have an explanation for the thin layer of cooling at a height of 13 km?
- Page 10, line 5: "other two in this figure". Do you mean three?
- Page 11, section 4.2: Would difference plots be better here, rather than showing the fields in both the original and modified runs. It would be easier for the reader to spot how the modified simulation differs, as comparing the figures is not so easy to do.

**References:**

Haynes P, McIntyre ME. 1987: On the evolution of vorticity and potential vorticity in the presence of diabatic heating and frictional or other forces. J. Atmos. Sci. 44: 828–841.

Kilroy G, Smith RK and Montgomery MT, 2018: The role of heating and cooling associated with ice processes on tropical cyclogenesis and intensification. Quart. J. Roy. Meteor. Soc. 144, 99–114.

---

## Referee Comment (RC2) · Anonymous Referee #2 · 28 Jun 2018

The manuscript presents a model study on the role of stratiform components of diabatic forcing in the development of a mid-level vortex. The authors implement the diagnosed diabatic forcing term into the model RAMS to examine the dynamic response to imposed latent heating and cooling. By several model experiments, they show that sublimation cooling at the base of the stratiform ice region is the main factor that causes the strong mid-level vortex to develop. The results of the study are interesting and the paper should be a welcome addition to the current literature on cloud microphysics and

tropical cyclogenesis. I would recommend that the manuscript be accepted after minor revisions especially in presentation. Below are my questions/suggestions in detail.

Specific comments:

1. The model sensitivity tests of the study are fine and very useful. It might be more helpful for the reader to understand the model results and associated mechanisms if the simulation scenarios could be introduced more explicitly. Specifically, the authors might consider merging the last second paragraph (page 6 , line 4-8) and perhaps also some parts of the last third paragraph (page 5, line 12 – page 6, line 3) of the Introduction section into the Methodology section (page 8, line 5-7). Then, detailed simulation scenarios need to be listed (perhaps in a table or at least with numbering), and so does other forcing than diabatic heating and cooling described in the last paragraph of Sect. 3 (page 8, line 8-13).

2. The manuscript focuses on the effect of cloud microphysical processes on a mid-level vortex. However, the basic principles that link the microphysics and the dynamics are not sufficiently described in some cases. I can see the results shown in the figures, but I do have some difficulties to follow the clues of the story. The authors might consider reorganizing the subsections of the Results section including the figures according to the model simulation scenarios mentioned above. Although precipitation drag has been taken as another forcing earlier in Sect. 3, it is not fully discussed with a subsection or paragraph in the Results. If this forcing is not so important, it might be omitted in Sect. 3.

3. It might be helpful if the vertical sections of mixing rations of each hydrometer could be shown with the figures in Sect. 4.1, which gives a general overview of the reference simulation. For the first sentence of Sect. 4.2 (page 11, line 1), what does "the changes discussed in section 3" mean specifically? Few previous studies are referred and compared in the discussions except for one part (page 10, line 25-26).

Technical issues:

[Figure]

Page 2, line 31-32: For "the cold air minimum", do you mean the minimum temperature or the coldest air?

Page 3, line 4: For "cod" pool, do you mean cool pool?

Page 3, line 9: "try and" can be omitted.

Page 3, line 21-22: The literature "Raymond et al. 2011" and "Gjorgjievska and Raymond 2013" cannot be found in the References list.

Page 3, line 11: "of Atlantic systems" should appear in front of "by Davis and Ahijevych (2012)".

Page 3, line 13: It should be "Bister and Emanuel (1997)".

Page 4, line 17-19: This sentence is vague and too long and it should be rephrased.

Page 4, line 22-24: What observational studies are needed, and are they available at present time? What do you mean by stating "that will emerge in the future"?

Page 4, line 27: Better to use "Many MCSs".

Page 4, line 32: Should be "for them to be".

Page 5, line 4: Should be "found that"?

Page 5, line 15: Should be "For this purpose".

Page 5, line 29-30: Do you mean reasonable agreement between the balanced model and the cloud model? It should be rephrased.

---

## Author Comment (AC1) · 4 Jul 2018

1. "The method employed by the authors to isolate the effects of the diabatic heating/cooling profiles seems particularly messy. I wonder whether it is the "best approach". The authors suggest that the vortex will be "not too different from the quasi-balanced state ..." although there appears to me to be lots of arbitrary decisions made to achieve this state. Is it not simpler and cleaner to take the averaged diabatic heating

profiles and run the Sawyer-Eliassen model with a balanced vortex?"

Yes, one approach would be to run the Sawyer-Eliassen model, however, that does involve making approximations to the governing equations. The conclusion that the stratiform heating components are not able to fully account for the mid-level inflow may not have been so obvious with the Sawyer-Eliassen model approach. Moreover the balanced state can be used to examine the hypothesis that the top-heavy heating profile of the convective cells in their late stage of development can contribute to the mid-level inflow. For instance, in the second part of this study an experiment is run where multiple cells are initiated with low-level warm bubbles. Then it can be seen that a weak mid-level inflow develops prior to a stratiform region forming aloft. This indicates the diabatic heating in the convective cells is playing a role in causing the mid-level inflow.

2. "In the introduction (page 3, lines 7-14) the authors describe a theory where a mid-level vortex descends to the surface. A vortex descending to the surface would violate Haynes and McIntyre (1987). There can be no net downward transport of vorticity from the middle level vortex, ruling out the possibility of genesis being a "top down" process. In any case, down flow from mid-levels would produce outflow near the surface, leading to a dilution of the vertical vorticity transported downward."

Here we are summarizing the results of Bister and Emanuel (1997) in regards to the descent of the vortex. In their words "After some time, the midlevel vortex expands downward toward the boundary layer". Although the mechanism responsible is perhaps not clear from their discussion we see no reason to discount their numerical modeling results. It is a theory that has been put forward and we think it should be mentioned in the introduction. While we agree that according to Haynes and McIntyre (1987) there can be no net downward transport of vorticity from the middle level vortex, there could be a lowering of the diabatic cooling with time, which could cause the inflow to become lower. This would be expected to cause spin up of the winds at lower levels and therefore give the appearance of a descent of the vortex. It is possible that

as sublimation and evaporation leads to increased humidity at the base of the developing mid-level vortex that this increased humidity reduces subsequent sublimation and evaporation of hydometeors falling from aloft and therefore more hydrometeors reach lower levels, thereby shifting the level of maximum diabatic cooling to lower levels with time. However this is a hypothesis and we are not seeing significant descent of the mid-level vortex in this early 12-hour period.

3. "Also in the introduction (page 3, lines 20-24) the authors introduce the theory that the mid-level vortex is conducive to convection with a more bottom heavy mass profile. A recent paper by Kilroy etal. 2018 found the opposite: namely, a simulation with warm-rain-only microphysics did not develop a mid-level vortex, but the convection had a bottom heavy mass flux profile. The simulation with ice-microphysics developed a mid-level vortex, but did not have a bottom heavy mass flux profile (see their Fig. 13)."

This is a theory that has been put forward by other researchers and there is some observational support for it, so we think it should be mentioned in the introduction. We appreciate the reviewer drawing attention to this new modeling result and it is probably worth adding a sentence pointing out the implications for this theory.

4. Page 4, lines 17-19: "Overall we consider that there is ample evidence that mid-level vortices may at times be playing an active role in tropical cyclogenesis, and even though they are unlikely to always be essential to the process, there are going to be consequences when a mid-level vortex develops which need to be further investigated" This sentence is very vague and confusing. How can there by ample evidence that it may at times play an active role? What is the active role a mid-level vortex plays? What are the consequences? Unlikely to always be essential, is there any evidence they are ever essential? I won't argue that mid-level vortices occur in numerical models (and in nature), but is their formation a consequence of ice microphysics (or other model parameters) and not a necessity for genesis?"

There are many observational studies that find mid-level vortices develop prior to tropical cyclogenesis. Our simulations with vertical wind shear suggest that there can be ramifications for the further development of the system once a prominent mid-level vortex forms. The mid-level vortex tends to be moist compared to the surroundings since it is a region where considerable sublimation and evaporation has occurred. Moreover we find a cooling anomaly centered near the melting level, which can be expected to reduce low-level stability. Both factors could favor new convective development. In an environment with vertical wind shear a misalignment occurs between the low-level and mid-level circulations. This misalignment results in a region of enhanced local shear between the centers of the low-level and mid-level circulations. This strong local shear could potentially influence the intensity and longevity of convective cells. It is possible that once a mid-level vortex has formed it could influence development a day or two later either through merger with other mid-level vortices or by providing an environment favorable for convective development. Since we are not discussing these simulations in this paper we should probably reword this sentence, although we do think that the role of mid-level vortices in tropical cyclogenesis is worthy of further investigation.

5. "On that note, have the authors considered using/redoing one of their simulations from NM13 which followed Pathway 1 (and included ice-microphysics) and doing a similar analysis? It would be interesting to understand why a similar simulation with ice did not develop a mid-level vortex. Surely these simulations would also contain ample mid-level cooling from sublimation."

Most of our simulations with ice develop a mid-level vortex, but in some cases they are not very strong and genesis follows Pathway 1. This could be sensitive to several factors that have yet to be fully explored. For instance if the initial environmental sounding is made warmer aloft then convective cells are less deep and the ice layer aloft is not so extensive or thick, leading to a weaker vortex.

6. "Start of section 4. I found the results section a little difficult to follow at first, as there was little to no description of the vortex evolution. I have no feel for vortex strength or development. There should be a better segue into the main results. Perhaps the

authors could think about including a plot showing the time evolution of the maximum tangential winds, radial winds, etc. and give a description of the vortex evolution. Another section that would be greatly improved by proper introduction is from line 19, page 10. Why the jump from a system analysis to describing a single cell in detail. Why is this cell in particular important? I wasn't sure of the significance of these results."

We shall take into consideration this comment. We thought it was important to show a convective cell in detail although we agree there could be a better introduction. This particular cell was chosen since it was isolated and so less impacted by strong circulations induced by nearby cells.

Minor comments:

1. Page 3, line 4. "cod pool"

Will correct.

2. Page 4, lines 12-13. "Also Hurricane Guillermo (1991) in the Eastern Pacific by Bister and Emanuel...". Perhaps you can rephrase this, it reads like Bister and Emanuel were responsible for the hurricane.

Will rephrase.

3. Page 7, line 13. "conservative" should be "conserved". 
 
 

Will correct.

4. Page 8, line 31 and page 9, line 14: I dislike the phrase "descending inflow". The plot is not showing any vertical motion. Do you mean that the height of the inflow is decreasing with decreasing radius?

Yes, this is a more correct statement.

5. Page 9, line 1: "This mid-level maximum of tangential winds appears to be associated with the midlevel inflow". Why "appears" to be, can there be any other reason?

[Figure]

Yes, this statement could be more decisive.

6. Page 9, lines 4-5. Can this low level cooling be from the initial vortex, even after 48 h has passed?

Yes, the signature of the initial vortex is still quite strong even at 48 h.

7. Page 9, line 6. Do you have an explanation for the thin layer of cooling at a height of 13 km?

It appears to be partly due to the stratiform diabatic heating aloft (Fig. 14d). Possibly it is also partly due to cells overshooting their equilibrium buoyancy level causing some adiabatic cooling. It is an interesting feature, but we don't have a good theoretical explanation at this time.

8. Page 10, line 5: "other two in this figure". Do you mean three?

This sentence will be reworded.

9. Page 11, section 4.2: Would difference plots be better here, rather than showing the fields in both the original and modified runs. It would be easier for the reader to spot how the modified simulation differs, as comparing the figures is not so easy to do.

We will carefully consider this recommendation.

---

## Author Comment (AC2) · 4 Jul 2018

Specific comments:

"1. The model sensitivity tests of the study are fine and very useful. It might be more helpful for the reader to understand the model results and associated mechanisms if the simulation scenarios could be introduced more explicitly. Specifically, the authors might consider merging the last second paragraph (page 6, line 4-8) and perhaps also

some parts of the last third paragraph (page 5, line 12 – page 6, line 3) of the Introduction section into the Methodology section (page 8, line 5-7). Then, detailed simulation scenarios need to be listed (perhaps in a table or at least with numbering), and so does other forcing than diabatic heating and cooling described in the last paragraph of Sect. 3 (page 8, line 8-13)."

These are some useful suggestions that we will definitely consider.

"2. The manuscript focuses on the effect of cloud microphysical processes on a mid-level vortex. However, the basic principles that link the microphysics and the dynamics are not sufficiently described in some cases. I can see the results shown in the figures, but I do have some difficulties to follow the clues of the story. The authors might consider reorganizing the subsections of the Results section including the figures according to the model simulation scenarios mentioned above. Although precipitation drag has been taken as another forcing earlier in Sect. 3, it is not fully discussed with a subsection or paragraph in the Results. If this forcing is not so important, it might be omitted in Sect. 3."

We will carefully consider these recommendations for reorganizing the subsections. We thought precipitation drag should at least be examined to see if it was important. Maybe there is a better way of saying that the issue of precipitation drag has been investigated.

"3. It might be helpful if the vertical sections of mixing rations of each hydrometer could be shown with the figures in Sect. 4.1, which gives a general overview of the reference simulation. For the first sentence of Sect. 4.2 (page 11, line 1), what does "the changes discussed in section 3" mean specifically? Few previous studies are referred and compared in the discussions except for one part (page 10, line 25-26)."

We will consider the suggestion of including the mixing ratios of each hydrometeor. We have shown the aggregates in Fig. 6a and cloud water and rain mixing ratio for a convective cell in Fog. 9, but it might help to add another figure. By changes we mean

the method to achieve a near gradient balanced state. This could be reworded. We will see if there are other previous studies that are relevant to this work that can be referred to.

Technical issues:

"Page 2, line 31-32: For "the cold air minimum", do you mean the minimum temperature or the coldest air?"

This could be reworded to be clearer.

"Page 3, line 4: For "cod" pool, do you mean cool pool?

Thank you for catching this error.

"Page 3, line 9: "try and" can be omitted."

Yes, this would be better.

"Page 3, line 21-22: The literature "Raymond et al. 2011" and "Gjorgjievska and Raymond 2013" cannot be found in the References list."

Thank you for noticing these errors.

Page 3, line 11: "of Atlantic systems" should appear in front of "by Davis and Ahijevych (2012)".

Yes, this would be an improvement.

"Page 3, line 13: It should be "Bister and Emanuel (1997)"."

This will be corrected.

"Page 4, line 17-19: This sentence is vague and too long and it should be rephrased."

We will rephrase this sentence.

"Page 4, line 22-24: What observational studies are needed, and are they available at
Interactive comment

[Figure]

time? What do you mean by stating "that will emerge in the future"?"

Well we think further observational studies would help to clarify the physical processes causing mid-level vortices to form and their role in tropical cyclogenesis. A statement like this might fit better in the conclusions section where it could be elaborated on.

"Page 4, line 27: Better to use "Many MCSs"."

Yes, this is better.

"Page 4, line 32: Should be "for them to be"."

Thank you for noticing the error.

"Page 5, line 4: Should be "found that"?"

We will make this correction.

"Page 5, line 15: Should be "For this purpose"."

We will make this change.

"Page 5, line 29-30: Do you mean reasonable agreement between the balanced model cloud model? It should be rephrased."

This sentence will be rephrased.

---

## Author Response (AR1)

Reply to reviewer#1

1. "The method employed by the authors to isolate the effects of the diabatic heating/cooling profiles seems particularly messy. I wonder whether it is the "best approach". The authors suggest that the vortex will be "not too different from the quasi-balanced state ..." although there appears to me to be lots of arbitrary decisions made to achieve this state. Is it not simpler and cleaner to take the averaged diabatic heating profiles and run the Sawyer-Eliassen model with a balanced vortex?"

Another approach would be to run the Sawyer-Eliassen model, however, that does involve making approximations to the governing equations. The conclusion that the stratiform heating components are not able to fully account for the mid-level inflow may not have been so obvious with the Sawyer-Eliassen model approach. Moreover the balanced state can be used to examine the hypothesis that the top-heavy heating profile of the convective cells in their late stage of development can contribute to the mid-level inflow. For instance, in the second part of this study an experiment is run where multiple cells are initiated with low-level warm bubbles. Then it can be seen that a weak mid-level inflow develops prior to a stratiform region forming aloft. This indicates the diabatic heating in the convective cells is playing a role in causing the mid-level inflow.

2. "In the introduction (page 3, lines 7-14) the authors describe a theory where a mid-level vortex descends to the surface. A vortex descending to the surface would violate Haynes and McIntyre (1987). There can be no net downward transport of vorticity from the middle level vortex, ruling out the possibility of genesis being a ``top down'' process. In any case, down flow from mid-levels would produce outflow near the surface, leading to a dilution of the vertical vorticity transported downward."

We have expanded the paragraph discussing this theory (page 3, line 7-26). We note that this theory of descent of the mid-level vortex to near the surface does not occur in recent cloud-resolving tropical cyclogenesis simulations. However the mid-level vortex does typically build downward with time in the simulations of NM13 by 1-2 km and we suggest a plausible mechanism based on the level of the peak diabatic cooling lowering with time. We also mention that vertical vorticity of the mid-level vortex is not transported downwards according to Haynes and McIntyre (1987).

3. "Also in the introduction (page 3, lines 20-24) the authors introduce the theory that the mid-level vortex is conducive to convection with a more bottom heavy mass profile. A recent paper by Kilroy *etal*. 2018 found the opposite: namely, a simulation with warm-rain-only microphysics did not develop a mid-level vortex, but the convection had a bottom heavy mass flux profile. The simulation with ice-microphysics developed a mid-level vortex, but did not have a bottom heavy mass flux profile (see their Fig. 13)."

We now include mention of the results of Kilroy *et al*. (2018) in this paragraph (page 4, line 1-3).

4. Page 4, lines 17-19: "Overall we consider that there is **ample evidence** that mid-level vortices **may at times** be playing an active role in tropical cyclogenesis, and even though

they are **unlikely to always be essential** to the process, there are going to be **consequences** when a mid-level vortex develops which need to be further investigated" This sentence is very vague and confusing. How can there by ample evidence that it may at times play an active role? What is the active role a mid-level vortex plays? What are the consequences? Unlikely to always be essential, is there any evidence they are ever essential? I won't argue that mid-level vortices occur in numerical models (and in nature), but is their formation a consequence of ice microphysics (or other model parameters) and not a necessity for genesis?"

We have added a detailed discussion of this topic on page 4 line 26 to page 5 line 24.

5. "On that note, have the authors considered using/redoing one of their simulations from NM13 which followed Pathway 1 (and included ice-microphysics) and doing a similar analysis? It would be interesting to understand why a similar simulation with ice did not develop a mid-level vortex. Surely these simulations would also contain ample mid-level cooling from sublimation."

Most of our simulations with ice develop a mid-level vortex, but in some cases they are not very strong and genesis follows Pathway 1. One of the factors seemed to be sea surface temperature with colder values producing less ice and a weaker mid-level vortex since the sublimation cooling was less. Warmer environmental temperatures aloft could also potentially reduce the amount of ice produced.

6. "Start of section 4. I found the results section a little difficult to follow at first, as there was little to no description of the vortex evolution. I have no feel for vortex strength or development. There should be a better segue into the main results. Perhaps the authors could think about including a plot showing the time evolution of the maximum tangential winds, radial winds, etc. and give a description of the vortex evolution. Another section that would be greatly improved by proper introduction is from line 19, page 10. Why the jump from a system analysis to describing a single cell in detail. Why is this cell in particular important? I wasn't sure of the significance of these results."

We have included another figure (Figure 1) with time series that describes the evolution of the system in more detail. We now give a better introduction to the discussion of a convective cell (page 1, line 30). This particular cell was chosen since it was isolated and so less impacted by strong circulations induced by nearby cells (page 11, line 34).

**Minor comments:**

1. Page 3, line 4. "cod pool"

Corrected.

2. Page 4, lines 12-13. "Also Hurricane Guillermo (1991) in the Eastern Pacific by Bister and Emanuel...". Perhaps you can rephrase this, it reads like Bister and Emanuel were responsible for the hurricane.

Added "investigated" (page 4, line 24).

3. Page 7, line 13. "conservative" should be "conserved". · · · · ·

Have corrected.

4. Page 8, line 31 and page 9, line 14: I dislike the phrase "descending inflow". The plot is not showing any vertical motion. Do you mean that the height of the inflow is decreasing with decreasing radius?

We have changed this sentence (page 10, line 7).

5. Page 9, line 1: "This mid-level maximum of tangential winds appears to be associated with the midlevel inflow". Why "appears" to be, can there be any other reason?

We have made this statement more decisive (page 10, line 10). However, there could be vertical transports of momentum by convective cells leading to increases in the tangential winds, and indeed we thing that is going on in this simulation further aloft. To be consistent with Haynes and McIntyre presumably there would be no net increase of vorticity in a layer, so perhaps the vorticity would increase inwards of the increased tangential winds and decrease outwards.

6. Page 9, lines 4-5. Can this low level cooling be from the initial vortex, even after 48 h has passed?

Yes, the signature of the initial vortex is still quite strong even at 48 h. We now state this more explicitly (page 10, line 14).

7. Page 9, line 6. Do you have an explanation for the thin layer of cooling at a height of 13 km?

It appears to be partly due to the stratiform diabatic heating aloft (Fig. 14d). Possibly it is also partly due to cells overshooting their equilibrium buoyancy level causing some adiabatic cooling. It is an interesting feature, but we don't have a good theoretical explanation at this time.

8. Page 10, line 5: "other two in this figure". Do you mean three?

This sentence has been reworded (page 11, line 15).

9. Page 11, section 4.2: Would difference plots be better here, rather than showing the fields in both the original and modified runs. It would be easier for the reader to spot how the modified simulation differs, as comparing the figures is not so easy to do.

Thank you for the suggestion. That is another way of presenting the results, but we think this way is reasonable.

Reply to reviewer#2

Specific comments:

"1. The model sensitivity tests of the study are fine and very useful. It might be more helpful for the reader to understand the model results and associated mechanisms if the simulation scenarios could be introduced more explicitly. Specifically, the authors might consider merging the last second paragraph (page 6, line 4-8) and perhaps also some parts of the last third paragraph (page 5, line 12 – page 6, line 3) of the Introduction section into the Methodology section (page 8, line 5-7). Then, detailed simulation scenarios need to be listed (perhaps in a table or at least with numbering), and so does other forcing than diabatic heating and cooling described in the last paragraph of Sect. 3 (page 8, line 8-13)."

We have moved much of the discussion on the procedure to create a near gradient wind balanced vortex to the methodology section (page 8, line 16 to page 9, line 12). We now include a table listing the experiments (page 21).

"2. The manuscript focuses on the effect of cloud microphysical processes on a mid-level vortex. However, the basic principles that link the microphysics and the dynamics are not sufficiently described in some cases. I can see the results shown in the figures, but I do have some difficulties to follow the clues of the story. The authors might consider reorganizing the subsections of the Results section including the figures according to the model simulation scenarios mentioned above. Although precipitation drag has been taken as another forcing earlier in Sect. 3, it is not fully discussed with a subsection or paragraph in the Results. If this forcing is not so important, it might be omitted in Sect. 3."

At the end of section 4.1 we have added a discussion that makes it clearer how the microphysical processes relate to the diabatic forcing (page 12, line 31). We have discussed in more detail the precipitation drag result (page 15, line 22).

"3. It might be helpful if the vertical sections of mixing rations of each hydrometer could be shown with the figures in Sect. 4.1, which gives a general overview of the reference simulation. For the first sentence of Sect. 4.2 (page 11, line 1), what does "the changes discussed in section 3" mean specifically? Few previous studies are referred and compared in the discussions except for one part (page 10, line 25-26)."

We have included additional panels to Fig. 7 (page 28) showing the hydrometeor mixing ratios in more detail. The sentence with "the changes" has been reworded (page 13, line 19). We include a discussion of relative humidity and refer to previous studies by Nolan (2007) and Kilroy et al. (2018), in section 4.1 (page 12, line 21-30).

Technical issues:

"Page 2, line 31-32: For "the cold air minimum", do you mean the minimum temperature or the coldest air?"

This has been changed to "cold anomaly" (page 2, line 32).

"Page 3, line 4: For "cod" pool, do you mean cool pool?

Thank you for catching this error.

"Page 3, line 9: "try and" can be omitted."

Yes, this change has been made.

"Page 3, line 21-22: The literature "Raymond et al. 2011" and "Gjorgjievska and Raymond 2013" cannot be found in the References list."

We have corrected these references.

Page 3, line 11: "of Atlantic systems" should appear in front of "by Davis and Ahijevych (2012)".

This has been changed.

"Page 3, line 13: It should be "Bister and Emanuel (1997)"."

This has been corrected.

"Page 4, line 17-19: This sentence is vague and too long and it should be rephrased."

We have included more discussion of the relevance of mid-level vortices (page 4, line 26-page 5, line 18).

"Page 4, line 22-24: What observational studies are needed, and are they available at time? What do you mean by stating "that will emerge in the future"?"

This sentence has been rephrased (page 5, line 22).

 "Page 4, line 27: Better to use "Many MCSs"."

This change has been made.

"Page 4, line 32: Should be "for them to be"."

Thank you for noticing this error.

"Page 5, line 4: Should be "found that"?"

We have corrected this sentence.

"Page have made this change

"Page 5, line 29-30: Do you mean reasonable agreement between the balanced model cloud model? It should be rephrased."

We think this sentence is okay, unless we are missing something. It is now in the methodology section (page 8, line 29).

[revised manuscript text omitted]